



# Composition and mass size distribution of nitrated and oxygenated aromatic compounds in ambient particulate matter from southern and central Europe – implications for origin

Zoran Kitanovski[1*], Pourya Shahpoury[1,2], Constantini Samara[3], Aristeidis Voliotis[3,4], Gerhard Lammel[1,5]

[1] Max Planck Institute for Chemistry, Multiphase Chemistry Department, Mainz, Germany

[2] Environment and Climate Change Canada, Air Quality Processes Research Section, Toronto, Canada

[3] Aristotle University of Thessaloniki, Department of Chemistry, Environmental Pollution Control Laboratory, Thessaloniki, Greece

[4] University of Manchester, School of Earth and Environmental Sciences, Centre for Atmospheric Sciences, Manchester, United Kingdom

[5] Masaryk University, Research Centre for Toxic Compounds in the Environment, Brno, Czech Republic

[*]now at: Lek Pharmaceuticals d.d., Ljubljana, Slovenia

Correspondence to: Zoran Kitanovski (z.kitanovski@mpic.de); Pourya Shahpoury (p.shahpoury@mpic.de)

**Abstract**
Nitro-monoaromatic hydrocarbons (NMAHs), such as nitrocatechols, nitrophenols and
nitrosalicylic acids, are important constituents of atmospheric particulate matter (PM) water
soluble organic carbon (WSOC) and humic-like substances (HULIS). Nitrated and oxygenated
derivatives of polycyclic aromatic hydrocarbons (NPAHs, OPAHs) are toxic and ubiquitous in the
ambient air; due to their light absorption properties, together with NMAHs they are part of aerosol
brown carbon (BrC). We investigated the winter concentrations of these substance classes in size-
resolved particulate matter (PM) from two urban sites in central and southern Europe, i.e. Mainz
(MZ), Germany and Thessaloniki (TK), Greece. $\sum_{11}$NMAH concentrations in PM$_{10}$ and total PM
were 0.51-8.38 and 12.1-72.1 ng m$^{-3}$ at MZ and TK site, respectively, whereas $\sum_8$OPAHs were 47-



1636 and 858-4306 pg m$^{-3}$, and $\sum_{17}$NPAHs were $\leq$90 and 76-578 pg m$^{-3}$, respectively. NMAHs
and the water-soluble OPAHs contributed 0.4 and 1.8%, and 0.0001 and 0.0002 % to the HULIS
mass, at MZ and TK, respectively. The mass size distributions of the individual substances
generally peaked in the smallest or second smallest size fraction i.e., <0.49 µm or 0.49-0.95 µm.
The mass median diameter (MMD) of NMAHs was 0.10 µm and 0.27 µm at MZ and TK,
respectively, while the MMDs of NPAHs and OPAHs were both 0.06 µm at MZ, and 0.12 and 0.10
µm at TK. Correlation analysis between NMAHs, NPAHs and OPAHs from one side and WSOC,
HULIS, nitrate, sulphate and potassium cation (K$^+$) from another, suggested that the fresh biomass
burning emissions dominated at the TK site, while aged air masses (influenced by biomass and
fossil fuel burning) were predominant at the MZ site.

## 1. Introduction

Atmospheric humic-like substances (HULIS) represent a complex mixture of aliphatic and
aromatic compounds with multiple functional groups, such as hydroxyl, carbonyl, carboxyl, nitro,
nitrooxy, and sulphate groups (Havers et al., 1998; Graber and Rudich, 2006; Hallquist et al., 2009;
Claeys et al., 2012). They are a major constituent of aerosol water-soluble organic carbon (WSOC),
contributing between 9 and 72% of WSOC mass (Decesari et al., 2000; Graber and Rudich, 2006;
Lin et al., 2010; Zheng et al., 2013). The distribution of HULIS molecular weights (MWs) is
unimodal and ranges between 100 and 500 Da with most of the compounds grouping around 200
Da (Graber and Rudich, 2006; Claeys et al., 2012; Song et al., 2018), unlike soil humic and fulvic
acids with MW distributions extending well beyond 1000 Da. Due to the presence of light-
absorbing polyconjugated and aromatic compounds (Duarte et al., 2005; Graber and Rudich, 2006;
Claeys et al., 2012; Zheng et al., 2013), HULIS are an important constituent of aerosol water-
soluble brown carbon (BrC; Laskin et al., 2015, and references therein). The intense light-
absorption of HULIS in the ultraviolet and violet and blue visible regions, between 200 and 500
nm, can affect aerosol optical properties and atmospheric photochemical processes (Andreae and
Gelencser, 2006). Owing to the presence of highly polar polyfunctional material, HULIS has
surface-active properties and can make aerosols act as cloud condensation nuclei (CCN). In the
aerosol aqueous phase, HULIS can increase the solubility of hydrophobic organic compounds and
change the reactivity and solubility of metal aerosols, owing to metal-complexation properties
(Graber and Rudich, 2006). Finally, due to the presence of redox-active moieties, HULIS can



catalyse electron transfer reactions and formation of reactive oxygen species (ROS), which could
pose oxidative stress in humans upon inhalation (Verma et al., 2015).
Biomass burning (BB) is considered as one of the main sources of HULIS in the atmosphere (Lin
et al., 2010; Claeys et al., 2012; Pavlovic and Hopke, 2012; Zheng et al., 2013) and an important
source of aerosol nitroaromatic compounds (NACs; Claeys et al., 2012; Song et al., 2018). Recent
studies found that nitro-monoaromatic hydrocarbons (NMAHs), such as 4-nitrocatechol (4-NC;
MW: 155 Da) and isomeric methyl-nitrocatechols (MNCs; MW: 169 Da) are abundant constituents
of particulate matter (PM) HULIS, originating from BB (Claeys et al., 2012; Song et al., 2018).
NMAHs are emitted into the atmosphere by primary and secondary processes. 4-NC, MNCs,
nitroguaiacols (NGs) and nitrosalicylic acids (NSAs) are predominantly formed by secondary
oxidation of lignin thermal decomposition products (m-cresol, phenols, methoxyphenols,
catechols, salicylic acid etc.) in the gas- and aqueous phase (Iinuma et al., 2010; Kelly et al., 2010;
Kroflič et al., 2015; Frka et al., 2016; Teich et al., 2017; Finewax et al., 2018; Xie et al., 2017;
Wang et al., 2019). Therefore, the yellow-coloured water-soluble 4-NC and MNCs have been
proposed as suitable tracers for highly oxidized secondary BB aerosols (Iinuma et al., 2010;
Kitanovski et al., 2012b; Kahnt et al., 2013; Caumo et al., 2016; Chow et al., 2016). In the past
decade, the ambient PM nitrocatechols (NCs) have been measured in several studies world-wide,
i.e. Europe (Iinuma et al., 2010; Zhang et al., 2010; Kitanovski et al., 2012b; Kahnt et al., 2013;
Mohr et al., 2013; Teich et al., 2014; Frka et al., 2016), South America (Claeys et al., 2012; Caumo
et al., 2016), North America (al Naiema and Stone, 2017), Asia (Chow et al., 2016; Li et al., 2016;
Wang et al., 2019) and Australia (Iinuma et al., 2016). They represent a significant fraction of the
PM organic carbon (OC), e.g. 0.8% in winter $PM_{10}$ collected at an urban background location in
Slovenia (range 0.4-1.3%; Kitanovski et al., 2012b), 0.75% in winter $PM_{10}$ collected at rural site
in Belgium (Kahnt et al., 2013) and ≈0.3% in $PM_{10}$ collected in Brazil during the BB season
(Caumo et al., 2016). Nitrosalicylic acids (2-hydroxy-nitrobenzoic acids) have been reported in
PM samples collected at rural (van Pinxteren and Herrmann, 2007; van Pinxteren et al., 2012; Teich
et al., 2017; Wang et al., 2018), urban (Kitanovski et al., 2012a and 2012b; Teich et al., 2017;
Wang et al., 2018) and remote (Wang et al., 2018) sites. Similar to NCs, they are mainly associated
with secondary BB aerosols (Kitanovski et al., 2012b; Teich at el., 2017; Wang et al., 2018).
Nitrophenols (NPs), structurally related compounds to NCs, are emitted from primary sources (e.g.
traffic, coal and wood combustion, industry and agricultural use of pesticides), which usually



predominate their secondary formation, especially in urban areas (Harrison et al., 2005; Cecinato
et al., 2005; Hoffmann et al., 2007; Iinuma et al., 2007; Zhang et al., 2010; Ganranoo et al., 2010;
Özel et al., 2011; Mohr et al., 2013; Kitanovski et al., 2012a and 2012b; Inomata et al., 2015; Teich
et al., 2017; Wang et al., 2018).
Polycyclic aromatic hydrocarbons (PAHs) and their nitrated and oxygenated derivatives (NPAHs
and OPAHs), as well as hydroxy derivatives (OH-PAHs), are ubiquitous in the atmosphere
(Walgraeve et al., 2010; Lammel, 2015; Bandowe and Meusel, 2017; Shahpoury et al., 2018). They
are primarily emitted from incomplete combustion of fossil fuels (Zielinska et al., 2004;
Karavalakis et al., 2010; Pham et al., 2013; Inomata et al., 2015), wood, coal and biomass burning
(Ding et al., 2012; Shen et al., 2012, 2013a and 2013b; Huang et al., 2014; Vicente et al., 2016).
The PAH derivatives are secondarily formed by the reaction of parent PAHs with atmospheric
oxidants such as OH, $NO_x$ and $O_3$. Some NPAHs have distinct sources; for instance, 3-
nitrofluoranthene (3-NFLT) and 1-nitropyrene (1-NPYR) are specifically associated with
combustion sources, whereas 2-nitrofluoranthene (2-NFLT) and 2-nitropyrene (2-NPYR) are
produced through oxidation of their parent species in the atmosphere (Bandowe and Meusel, 2017).
Similarly, OPAHs benzanthrone (OBAT), benz(a)fluorenone (BaOFLN) and benz(b)fluorenone
(BbOFLN) have been associated with primary sources, whereas 9,10-anthraquinone (9,10-
$O_2$ANT), 1,2-benzanthraquinone (1,2-$O_2$BAA), and 9-fluorenone (9-OFLN) have been attributed
to both source types (Kojima et al., 2010; Souza et al., 2014; Lin et al., 2015; Zhuo et al., 2017).
The primary sources dominate in winter time with residential heating surpassing traffic emission
(Lin et al., 2015). It is anticipated that functionalized 2- and 3-ring PAHs (e.g. 2- and 3-ring
OPAHs) would exhibit the highest hydrophilicity among their analogs and could also be part of
PM HULIS (Vione et al., 2014; Fan et al., 2016; Haynes et al., 2019). The water-soluble OPAHs,
in particular quinones, were suggested to contribute to light-absorption properties of brown carbon
(Laskin et al., 2015; Haynes et al., 2019). Moreover, the ROS activity of HULIS from $PM_{2.5}$ was
associated to OPAHs, i.e. quinones and hydroxy-quinones (Verma et al., 2015). It has been shown
in controlled experiments that the chemical aging of PM from various origins would increase its
ROS activity and this effect is enhanced in the presence of $O_3$ (Li et al., 2009; McWhinny et al.,
2011; Stevanovic et al., 2013; Verma et al., 2014 and 2015; Antiñolo et al., 2015). This process
has been attributed to oxidation of PAHs and formation of water-soluble derivatives.



NMAHs, PAHs and N/OPAHs significantly contribute to the aerosol BrC due to their light-
absorption capacity in the UV and visible range (Mohr et al., 2013; Teich et al., 2017; Xie et al.,
2017). Determining the size-resolved mass distribution of the PM molecular tracers is important
for assessing the particle emission sources, atmospheric transport, and health effects (Neusüss et
al., 2000). In particular, there is a limited knowledge about the size-resolved characteristics of
NMAHs and N/OPAHs, and their relation to atmospheric HULIS (Claeys et al., 2012; Song et al.,
2018). Therefore, the aim of the present work is to fill this gap by studying the size-resolved PM
from polluted urban air at two locations in central and southern Europe, i.e. Mainz (MZ), Germany
and Thessaloniki (TK), Greece, and to apply these data to determine the possible emission sources.
The concentrations of ions, organic acids, HULIS and HULIS-C in the samples used in this study
can be found in a companion paper (Voliotis et al., 2017).

**2. Experimental**
**2.1 Chemicals and solutions**
Solvents including methanol (MeOH, Chromasolv, LC-MS grade; Fluka, Buchs, Switzerland),
tetrahydrofuran (THF, LiChrosolv, HPLC grade; Merck, Darmstadt, Germany), high-purity water
(18.2 MΩ cm; Elga PURELAB, Veolia Water Technologies, Celle, Germany),
ethylenediaminetetraacetic acid (EDTA, trace metals basis; Sigma-Aldrich, St. Louis, USA),
formic acid and ammonium formate (grade eluent additive for LC-MS; Fluka) were used for LC-
MS mobile phase and sample preparation for NMAHs. Dichloromethane (DCM), *n*-hexane, and
ethyl acetate (Suprasolv, GC-MS grade, Merck) were used for N/OPAH analysis. Analytical
standards used in our study, their acronyms, and suppliers are listed in Tables 1 and S1. The internal
standards (IS) of 2,4,6-trinitrophenol (picric acid, aqueous solution 1.0%; Sigma-Aldrich) and 4-
nitrophenol-$d_4$ (4-NP-$d_4$; LGC, Teddington, UK) were used for NMAH quantification, whereas 1-
nitronaphthalene-$d_7$,   2-nitrofluorene-$d_9$,   9-nitroanthracene-$d_9$,   3-nitrofluoranthene-$d_9$,   1-
nitropyrene-$d_9$, 6-nitrochrysene-$d_{11}$, 9,10-anthraquinone-$d_8$, and 9-fluorenone-$d_8$ (Chiron, Norway)
were used for N/OPAH quantification. Individual stock solutions of NMAH standards were
prepared in methanol at concentrations of 200 µg mL$^{-1}$, whereas those for N/OPAHs were prepared
in toluene at 10 µg mL$^{-1}$. Standard mixtures were prepared for each substance class from individual
stock solutions, and further used for preparation of calibration standards of NMAHs in
methanol/water mixture (3/7, v/v) containing 5 mM ammonium formate buffer pH 3 and 400 µM





EDTA (injection solvent), and calibration standards of N/OPAHs in ethyl acetate. NMAH and
N/OPAH calibration standards were prepared in the concentration range of 0.1 to 500 and 0.25 to
1000 pg $\mu L^{-1}$, with a fixed IS concentration of 100 and 200 pg $\mu L^{-1}$, respectively.

**2.2 Collection of samples**
All PM samples were collected using a 5-stage high-volume cascade impactor with effective cut-
off diameters: 0.49, 0.95, 1.5, 3 and 7.2 μm of aerodynamic particle size, $D_p$, and a backup filter
collecting particles < 0.49 μm (Table 2). The sampling in MZ was done using a high-volume air
sampler Baghirra HV-100P (Baghirra, Prague, Czech Republic) equipped with a multi-stage
cascade impactor (Tisch Environmetal Inc., Cleves, USA, series 230, model 235) and a $PM_{10}$ head.
Downstream of the impactor, gaseous organics were collected in two polyurethane foam plugs
(PUF; density 0.030 g $cm^{-3}$; Organika, Malbork, Poland) placed in a glass cartridge. The PM was
sampled on slotted quartz fibre filters (QFFs, TE-230-QZ, Tisch Environmental Inc., 14.3×13.7
cm) and a QFF backup filter (Whatman, 20.3×25.4 cm). Four sets of samples were collected at MZ
between November and December 2015, each over the period of 70 hrs (flow rate: 60 $m^3 h^{-1}$; Table
2). The impactor used in TK was a Sierra Instruments, model 235; the PM samples were collected
on QFFs (Tisch Environmental TE-230QZ, slotted 5.7×5.7 cm) and on QFF backup filters (Pall,
2500 QAT-UP), without a $PM_{10}$ head, as described in Voliotis et al. (2017).

**2.3 Sample preparation and analytical methods**
**2.3.1 LC/MS analysis of nitro-monoaromatic hydrocarbons**
Extraction of the filter samples for NMAH analysis was done using a validated procedure
(Kitanovski et al., 2012b) with small modifications. Briefly, a 1.5 $cm^2$ section of the filter was
spiked with both IS (spiked mass: 100 ng) and subsequently extracted three times (5 min each)
with 10 mL methanolic solution of EDTA (3.4 nmol $mL^{-1}$) in an ultrasonic bath. The combined
extracts were concentrated to 0.5 mL using a TurboVap II (bath temperature: 40°C, nitrogen gas
pressure: 15 psi; Biotage, Uppsala, Sweden). The concentrated extract was filtered through a 0.2-
μm PTFE syringe filter (4 mm, Whatman; GE Healthcare, Little Chalfont, UK) into a 2-mL vial
and was evaporated to near dryness under the gentle stream of nitrogen (99.999%; Westfalen AG,
Münster, Germany). Finally, the extract was dissolved in methanol/water mixture (3/7, *v/v*)
containing 5 mM ammonium formate buffer pH 3 and 400 μM EDTA for LC/MS analysis.



The NMAHs were determined using an Agilent 1200 Series HPLC system (Agilent Technologies,
Waldbronn, Germany) coupled to an Agilent 6130B Series single quadrupole mass spectrometer
equipped with an electrospray ionization (ESI) source. High-purity nitrogen was used as nebulizer
and drying gas. The separation of the targeted analytes was done on an Atlantis T3 column (150
mm × 2.1 mm i.d., 3 μm particles size; Waters, Milford, USA), connected to an Atlantis T3
VanGuard pre-column (5 mm × 2.1 mm i.d., 3 μm particles size; Waters), using isocratic elution
with a mobile phase consisted of MeOH/THF/water (30/15/55, $v/v/v$) mixture containing 5 mM
ammonium formate buffer pH 3. The mobile phase flow rate, column temperature and injection
volume were 0.2 mL min$^{-1}$, 30°C and 10 μL, respectively (Kitanovski et al., 2012b). The detection
and quantification of NMAHs was done in single ion monitoring and negative ESI mode (Table
1). The optimized ESI-MS parameters were as follows: –1000V for the ESI capillary voltage, 30
psig for the nebulizer pressure and 12 L min$^{-1}$ and 340°C for the drying gas flow and temperature,
respectively. Due to the lack of a reference standard for 3-methyl-4-nitrocatechol (3-M-4-NC), its
concentrations were calculated based on the calibration curve of 4-M-5-NC. This is justified based
on the structural similarity of the two substances and therefore similar ionization efficiency under
ESI-MS conditions. LC/MSD ChemStation (Agilent Technologies) was used for data acquisition
and analysis.

**2.3.2 Chemical analysis of nitro- and oxy-polycyclic aromatic hydrocarbons**
N/OPAHs were extracted from PM samples following a QuEChERS method with slight
modifications (Albinet et al., 2014; Shahpoury et al., 2018). Briefly, each filter paper was placed
inside a glass centrifuge tube (Duran, Schott, Mainz, Germany) and spiked with a mixture of
internal standards containing 60 ng of each 1-nitronaphthalene-d$_7$, 2-nitrofluorene-d$_9$, 9-
nitroanthracene-d$_9$, 3-nitrofluoranthene-d$_9$, 1-nitropyrene-d$_9$, 6-nitrochrysene-d$_{11}$, 9,10-
anthraquinone-d$_8$, and 9-fluorenone-d$_8$. 7 mL of DCM was then added to each tube, the tubes were
capped and the samples were extracted by vortexing for 1.5 min. The extracts were passed through
a glass funnel plugged with deactivated glass wool and concentrated to 0.5 mL using a TurboVap
II. The concentrated extracts were loaded on pre-conditioned SiO$_2$ solid-phase extraction cartridges
(500 mg; Macherey-Nagel, Weilmünster, Germany) and the target analytes were eluted with 9 mL
of 65:35 $n$-hexane-DCM.


The purified extracts containing the analytes were concentrated to 0.5 mL and the solvent was
exchanged by adding 5 mL of ethyl acetate, concentrating the solution to 0.5 mL, and repeating the
process three times. The sample volumes were adjusted to 0.3 mL and transferred to 2 mL vials
containing 0.4 mL glass inserts. All solvents used for N/OPAH analysis were high-purity
(Suprasolv, GC-MS; Merck, Darmstadt, Germany). All glassware used for analysis was pre-
washed with lab-grade detergent, tap water and deionized water, and baked at 310°C for 12 hours.
The samples were analysed using a Trace 1310 gas chromatograph (GC; Thermo Scientific,
Waltham, MA, USA) interfaced to a TSQ8000 Evo triple-quadrupole mass selective detector
(MS/MS; Thermo Scientific). The analysis was performed in negative chemical ionization with
methane used as ionization gas (1.5 mL min$^{-1}$ flow rate; > 99.99%; Messer, Bad Soden, Germany).
The analytes were separated on a 30-m DB-5ms capillary column (0.25 mm ID, 0.25 µm film
thickness; J&W, Santa Clara, CA, USA) with helium (99.99 %; Westfalen AG, Münster, Germany)
as carrier gas at 1 mL min$^{-1}$ flow rate. The GC inlet temperature was set to 250°C and operated in
pulsed splitless mode (30 psi pulsed pressure for 1.5 min, and splitless time of 1.8 min). The GC
oven temperature was held at 60°C for 2 min at the start of the analysis, then increased to 180°C at
15°C min$^{-1}$, and to 280°C at 5°C min$^{-1}$, followed by a final hold time of 15 min. MS transfer line
and ion source temperature were set to 290 and 230°C, respectively. Emission current and electron
energy were set to 100 µA and −70 eV, respectively. The target analytes were detected in selected
ion monitoring mode, identified using their retention times and quantification ions (Table 1). The
quantification was performed using the internal calibration method and 11-point calibration curves
ranging from 0.25 to 1000 pg µL$^{-1}$. Trace Finder (Thermo Scientific, Waltham, USA) was used
for chromatographic data acquisition and analysis.

### 218    2.3.3 Quality control and data analysis

Field blanks ($n$ = 3) were prepared during sample collection by mounting the pre-baked filters on
the sampler without switching it on. These filters were subsequently retrieved and processed along
with the rest of the samples. Limits of quantification (LOQ) for analytes were calculated as mean
concentration of each analyte in blanks + 3 standard deviations. When analyte concentrations in
the samples exceeded the LOQ, mean blank concentrations were subtracted from those in the
corresponding samples. Microsoft Office Excel 2013 (Microsoft Corp., Redmond, USA) and
OriginPro 9.0 (OriginLab Corp., Northampton, USA) were used for statistical analysis and data



visualization. Mass size distributions (MSDs) of NMAHs and N/OPAHs were additionally
characterized by the mass median diameter (MMD), defined as log MMD = $\Sigma$ ($c_i$ log $D_i$)/ $\Sigma$ $c_i$,
with $c_i$ and $D_i$ being the concentration (ng m$^{-3}$) and geometric mean diameter, respectively, of six
impactor stages. 0.001 μm was adopted as the lower cut-off of the lowermost stage (backup filter)
and 10 μm as the upper cut-off of the uppermost stage, even in the absence of a PM$_{10}$ head (i.e. TK
samples).

**3. Results and discussion**
**3.1 Levels of NMAHs**
From the 11 targeted NMAHs, 8 were consistently detected in size-segregated PM from MZ and
TK. 4-NG and DNOC were not detected in MZ samples, while being sporadically detected in the
coarse PM (>3 μm) from TK. 2,4-DNP was detected more frequently in TK (three sample sets)
than in MZ samples (one sample set).
The concentrations of NMAHs associated to PM$_{10}$ (MZ) and total PM (TK) are given in Table S3.
PM$_{10}$ and total PM $\sum_{11}$NMAH concentrations in MZ and TK were 0.51-8.38 and 12.1-72.1 ng m$^{-}$
$^3$, respectively. In all sample sets, 4-NC was the most abundant NMAH with concentrations ranging
within 0.05-3.90 ng m$^{-3}$ (mean 2.46 ng m$^{-3}$; Table S3) in MZ samples, and 10 times higher
concentrations in TK samples (5.89-36.33 ng m$^{-3}$; mean 22.11 ng m$^{-3}$; Table S3). Second most
abundant NMAH in MZ was found to be 4-NP with concentrations between 0.24 and 1.27 ng m$^{-3}$
(mean 0.83 ng m$^{-3}$; Table S3), while 4-M-5-NC was the second most abundant in TK samples (2.54
- 16.05 ng m$^{-3}$; mean: 9.79 ng m$^{-3}$; Table S3). In general, the concentration trends of NMAHs were
4-NC > MNCs > 4-NP > NPs > NSAs > DNP (dinitrophenols) for MZ samples, and 4-NC > MNCs
> 4-NP > NSAs > NPs > DNP for TK samples. These trends are in good agreement with other
studies, where 4-NC, MNCs and 4-NP were the most abundant NMAHs (Kitanovski et al., 2012b;
Chow et al., 2016). However, we previously found different concentration trends in snow-
scavenged atmospheric particles collected in MZ, where 4-NC and MNCs were the second most
abundant NMAH species following NPs (Shahpoury et al., 2018). $\sum$NMAH winter concentrations
at TK were higher than those found in winter PM$_{2.5}$ and PM$_{10}$ from Hong Kong (China; Chow et
al., 2016) and rural Belgium (Kahnt et al., 2013), respectively, but lower than NMAH
concentrations in winter PM$_{10}$ samples from Ljubljana (Slovenia; Kitanovski et al., 2012b) and
Shanghai (China; Li et al., 2016). The concentrations of individual NMAHs in winter PM$_{10}$ from





MZ were among the lowest values reported so far (Iinuma et al., 2010; Kitanovski et al., 2012b;
Kahnt et al., 2013; Mohr et al., 2013; Chow et al., 2016; Li et al., 2016; Teich et al., 2017; Wang
et al., 2019).
In Table S3, one can easily notice the consistently higher (≈10 times) total PM concentrations of
4-NC, MNCs and NSAs in TK samples compared to those found in $PM_{10}$ samples from MZ.
Smaller concentration discrepancies among the sites were observed for 4-NP and methyl-
nitrophenols (MNPs) (up to 3 times higher concentrations in TK samples). Since 4-NC, MNCs and
NSAs are considered as suitable tracers for BB aerosols (Iinuma et al., 2010; Kitanovski et al.,
2012b; Kahnt et al., 2013; Caumo et al., 2016; Chow et al., 2016; Teich et al., 2017), this suggests
that the air masses over TK during sample collection were greatly influenced by BB emissions. To
test this hypothesis, a correlation analysis was done for NMAHs. Except for NPs in TK samples,
generally high correlations were observed within the NMAHs compound groups (NSAs, NCs, NPs;
$R^2_{adj} > 0.8$; Table S4 and S5). The correlation analysis of TK samples showed several interesting
features (Table S4). Firstly, 5-NSA highly correlated ($R^2_{adj}$ 0.81 − 0.83) with 4-NP and potassium
cation ($K^+$), but showed insignificant correlations with 4-NC, MNCs and nitrate. Moreover, 3-NSA
showed significantly ($p<0.05$) high correlation only with $K^+$, but moderate with 4-NP.
Additionally, 4-NP was highly correlated with $K^+$ and nitrate ($R^2_{adj}$ 0.94 and 0.81, respectively).
Secondly, 4-NC and 4-M-5-NC showed low correlations with $K^+$ and nitrate, but highly correlated
with 3-M-4-NP ($R^2_{adj}$ 0.74 and 0.78, respectively). In our previous work (Kitanovski et al., 2012b),
high correlations between NSAs and 4-NC, MNCs or nitrates were observed ($R^2_{adj} > 0.8$),
supporting NSAs' secondary origin (Kitanovski et al., 2012b; Teich et al., 2017). Our TK results
indicate different emission sources between NSAs and 4-NP on the one hand, and 4-NC and MNCs
on the other hand. 5-NSA and 4-NP (2-M-4-NP included) most likely had the same emission
source, i.e. BB (both correlate with $K^+$), and were probably formed by aqueous-phase nitration of
their phenolic precursors (especially for 4-NP and 2-M-4-NP, which both highly correlated with
nitrates) in deliquescent aerosol (Kroflič et al., 2018). Additionally, 3-NSA (insignificantly
correlated with nitrate) was probably emitted primarily by BB (Wang et al., 2017). In contrast, low
correlations of 4-NC and MNCs with $K^+$ and nitrates suggest that BB and aqueous-phase nitration
might not be the dominating emission sources, and that their possible main source could be gas-
phase nitration of anthropogenic precursors (Finewax et al., 2018), such as benzene and toluene
(Xie et al., 2017; Wang et al., 2019), emitted from fossil fuel combustion (e.g. traffic, coal


combustion). The statistically significant (p<0.05) correlations of 3-M-4-NP with 4-NC and 4-M-
5-NC, in contrast to the correlations with 4-NP and nitrates, suggest that most likely 3-M-4-NP had
similar emission sources with NCs (i.e. fossil fuel combustion). It can be noted from Table S4 that
MNP isomers (2-M-4-NP and 3-M-4-NP) probably had different main emission sources, i.e.
aqueous-phase nitration of a 2-M-4-NP precursor emitted from BB *vs.* fossil fuel combustion in
case of 3-M-4-NP (Noguchi et al., 2007).
Correlation analysis for NMAHs in MZ samples presents quite a different picture (Table S5).
Statistically significant (p<0.05) high correlations were observed among different NMAH
compound groups (i.e. NSAs, NCs, NPs), with most of $R^2_{adj}$ higher than 0.8. $K^+$ was correlated
with 5-NSA, 4-NC, MNCs and 3-M-4-NP, indicating their predominant emission from BB. Nitrate
showed high correlations ($R^2_{adj}$>0.9) with 3-NSA, 4-NP, 2-M-4-NP and 2,4-DNP, suggesting that
aqueous-phase nitration was a main source for these compounds over MZ (Table S5). Two pairs
of positional isomers i.e. 3-NSA/5-NSA and 2-M-4-NP/3-M-4-NP showed distinct correlations
within their pair with regard to nitrate and $K^+$. 3-NSA and 2-M-4-NP, which were highly correlated
with nitrates, showed no correlation with $K^+$, indicating that aqueous-phase chemistry could have
played a significant role in their formation. In contrast, the opposite was observed for 5-NSA and
3-M-4-NP (Table S5). In summer $PM_{2.5}$ over a rural site in northern China, Wang et al. (2018)
observed weak correlations of NSAs with $NO_2$ that could indicate formation processes other than
nitration. Primary NSA emission from traffic or BB cannot be excluded, since their positional
isomers were found in diesel exhaust particles (Seki et al., 2010) or in BB smoke particles (Wang
et al., 2017). The correlations of 4-NC and MNCs with $K^+$, 4-NP and MNPs suggest similar sources
for NCs and NPs over MZ (Chow et al., 2016; Voliotis et al., 2017; Wang et al., 2018).

**3.2 Mass size distributions of NMAHs**
MSDs of NMAHs over the two sampling locations are given in Fig. 1 and 2. NSAs (3-NSA and 5-
NSA) and NCs (4-NC, 4-M-5-NC, 3-M-5-NC and 3-M-4-NC) showed unimodal distributions with
MSDs generally peaking in the finest PM fraction ($PM_{0.49}$) in both MZ and TK samples. Overall,
NMAHs were prominent in smaller size fractions ($PM_{0.95}$) in MZ compared to TK (Fig. 1 and 2).
For NSAs, in one out of the four samples collected at MZ, MSDs peaked in $PM_{1.5-0.95}$ fraction,
while the $PM_{0.95}$ mass fractions of 3-NSA and 5-NSA were 22% and 44%, respectively (Fig. S1a).
In this sample only, 5-NSA showed bimodal distribution (dominant peaks in $PM_{0.49}$ and $PM_{1.5-0.95}$).


Moreover, 4-NP and MNPs were the most abundant NMAHs (Fig.S1a). The dominant MSD peak
of NSAs in $PM_{1.5-0.95}$ and the concentration abundance of 4-NP and MNPs could indicate possible
influence of primary traffic emissions (fossil fuel combustion; Seki et al., 2010; Inomata et al.,
2015) at the beginning of the sampling campaign in MZ. During the next sampling periods at MZ
site (Figs. S1b, S1c and S1d), 75-86% of NSAs' $PM_{10}$ mass was associated with $PM_{0.95}$, which is
in line with the observations at TK (66-82% of total PM mass belongs to $PM_{0.95}$; Fig. S2). At both
sites, usually more than 90% of the compound total mass was associated with $PM_3$ (range: 83-
99%). 87-93% and 82-88% of NCs at MZ and TK were associated with $PM_{0.95}$ (Figs. S1, S2 and
S5). The coarse mode (>3 μm) accounted for only 1% (MZ) or 2.5% (TK). The larger coarse
fraction found in TK could be partially attributed to the fact that the sampling system did not have
a $PM_{10}$ inlet, thus it could potentially collect coarse particles up to approximately 30 μm (Voliotis
et al., 2017). The unimodal distributions of NCs peaking in the fine PM fraction are in line with
the only report on MSDs of 4-NC (Li et al., 2016). The MSDs of HULIS in MZ and TK closely
followed the MSDs of NCs and NSAs (Figs. 1 and 2), suggesting that these compounds could be
important constituents of PM HULIS (for detailed discussion see Sect. 3.5). The accumulation of
the NCs' and NSAs' mass in the submicrometer (<0.95 μm) PM fractions could indicate fresh
combustion emissions (e.g. BB) and/or gas-to-particle conversion processes of their precursors
over MZ and TK (Li et al., 2016).
Nitrophenols (i.e. 4-NP, 2-M-4-NP and 3-M-4-NP) showed bimodal distributions with a dominant
peak in the finest fraction ($PM_{0.49}$) and a smaller peak in $PM_{3-0.95}$ (Figs. 1, 2, S1, S2 and S5).
Bimodal distribution of NPs (i.e. 4-NP, 4-NG, 2,6-dimethyl-4-nitrophenol and 2,6-dinitrophenol)
with a small mode peak in the fine PM fraction and a big one in the coarse fraction, was recently
reported during winter haze episodes over Shanghai, China (Li et al., 2016). Our results imply that
BB and gas-to-particle conversion processes were likely more prevalent emission sources for NPs
in MZ and TK (dominant NPs' peak in $PM_{0.49}$) than fossil fuel (diesel) combustion sources
(Harrison et al., 2005; Noguchi et al., 2007; Inomata et al., 2015; Li et al., 2016). For 4-NP, at both
sites, around 80% of $PM_{10}$ mass (or of the total PM mass at TK) was associated with $PM_3$, while ≈
60% was associated with $PM_{0.95}$ (Figs. S1 and S2). Similarly, for methyl-nitrophenols 83-88% of
$PM_{10}$ mass at MZ and 75-83% of total PM mass at TK sites were associated with $PM_3$, while 58-
65% of $PM_{10}$ at MZ and 48-61% of total PM mass at TK sites were associated with $PM_{0.95}$ (Figs.
S1 and S2).



MMD of NMAHs was 0.10 µm (0.24 for NPs, 0.07 for NCs and 0.11 µm for NSAs) at MZ *vs*. 0.27
µm (0.60 for NPs, 0.24 for NCs and 0.31 µm for NSAs) at TK. The larger MMDs at TK could be
explained by the larger size range collected at this site as mentioned above, but they could also be
indicative of aerosol aging. In aged aerosols, semivolatiles are expected to be re-distributed with
the MMD approaching the surface mean diameter, which for urban and continental aerosol peaks
around 0.2 µm (Jaenicke, 1988), a shift which could not be resolved by the sampling technique
applied. Note that the low size resolution (6 stages) may hide modes, which in particular applies
for the so-called accumulation mode, which adds mostly to $PM_{0.49}$, but also to the size fraction
between 0.49 and 0.95 µm.

**3.3 Levels of N/OPAHs**
N/OPAHs were studied in size-resolved PM in both MZ and TK sites. At both sites, particle-phase
OPAHs were detected more frequently than NPAHs: seven out of eight OPAHs targeted for
analysis were detected in nearly all MZ and TK samples (Table S3; Figs. S3 and S4). In contrast,
only eight out of seventeen targeted NPAHs were found in the PM samples, of which only 1-
nitronaphthalene (1-NNAP), 9-nitroanthracene (9-NANT), 2-NFLT, and 7-nitrobenz(*a*)anthracene
(7-NBAA) were detected in both MZ and TK samples. Interestingly, 3-nitrophenanthrene (3-
NPHE), 3-NFLT, and 1- and 2-NPYR were only found in TK samples. This was not due to
differences in individual LOQs between the two sites (see Table S2). The mean concentrations of
NPAHs in PM were dominated by 9-NANT followed by 2-NFLT and 7-NBAA at both sites (Figs.
1 and 2, Table S3), with concentrations reaching to 225, 154, and 71 pg m$^{-3}$, respectively. This
pattern closely resembles those previously reported for PM from several locations in central Europe
(Tomaz et al., 2016, and references therein), including NPAHs found in snow-scavenged
atmospheric particles from MZ sample site (Shahpoury et al., 2018). As for OPAHs, the mean
analyte concentrations in PM were dominated by OBAT, followed closely by BbOFLN, BaOFLN,
9,10-O$_2$ANT, and 1,2-O$_2$BAA. The latter two quinones could be of high importance due to their
redox activity, and their potential to catalyse the formation of reactive oxygen species (ROS) within
the human respiratory system (Ayres et al., 2008; Bates et al., 2019). The two substances were
found to dominate two out of four MZ samples with concentrations up to 221 and 137 pg m$^{-3}$,
respectively. These concentrations were higher at TK site and reached 354 and 514 pg m$^{-3}$,
respectively.


Overall, all N/OPAHs showed considerably higher concentrations in TK than in MZ samples.
∑NPAH concentrations in $PM_{10}$ from MZ and in total PM from TK were <LOQ-90 and 76-578 pg
$m^{-3}$, respectively, whereas ∑OPAHs demonstrated much higher levels ranging 47-1636 and 858-
4306 pg $m^{-3}$, respectively. The sum of three quinones 1,4-naphthoquinone ($1,4-O_2NAP$), 9,10-
$O_2ANT$, and $1,2-O_2BAA$ were 30-363 and 428-873 pg $m^{-3}$, respectively. The levels of particle-
phase NPAHs found in MZ fall in the lower end of the range (50-500 pg $m^{-3}$) observed for various
types of sites in Europe (Tomaz et al., 2016, and references therein). The levels at TK represent the
upper end of this range, while being within the concentration range previously found at other sites
in Thessaloniki (1204 ± 249 pg $m^{-3}$ at a traffic site, 383 ± 77 pg $m^{-3}$ at an urban background site,
Besis et al., 2017). The total OPAH concentrations at both sites fall in the lower end of the range
previously observed in Europe (0.5-50 ng $m^{-3}$; Tomaz et al., 2016 and references therein).
N/OPAHs were predominant in the sub-micrometre PM fraction ($PM_{0.95}$; 85-91% of $PM_{10}$ at MZ
and 78-85% of total PM at TK site; Figs. 1, 2, S3, S4 and S5), with relatively more enrichment in
$PM_{0.49}$ compared to $PM_{0.49-0.95}$ across the two sites. The mean concentrations of ∑NPAHs in $PM_{0.49}$
from MZ and TK were 101±73 and 417±134 pg $m^{-3}$, whereas in $PM_{0.49-0.95}$ were 22.8±15.9 and
222±95 pg $m^{-3}$, respectively. ∑OPAHs showed similar patterns at MZ and TK sites – they were
460±566 and 1426±1210 pg $m^{-3}$ in $PM_{0.49}$, respectively, and 81.6±78.8 and 555±209 pg $m^{-3}$ in
$PM_{0.49-0.95}$. The targeted NPAHs did not show a second mode in any sample, whereas for 9-OFLN
and $9,10-O_2ANT$ a second mode was found in MZ samples. Such differences between mass
distributions indicate that these OPAHs are subject to different atmospheric processes compared to
the other N/OPAHs that we studied. This could point at different emission and formation pathways
in the atmosphere (see Sect. 3.4 for further discussion). Some of the OPAHs with single O-atom,
namely OBAT, BaOFLN, and BbOFLN, originate from primary sources (i.e. combustion of fossil
fuels and biomass; Albinet et al., 2007; Karavalakis et al., 2010; Shen et al., 2013b; Souza et al.,
2014; Huang et al., 2014; Tomaz et al., 2016; Vicente et al., 2016), whereas some quinones, such
as $9,10-O_2ANT$ and $1,2-O_2BAA$, are associated with both primary and secondary sources (Kojima
et al., 2010; Souza et al., 2014; Lin et al., 2015; Zhuo et al., 2017). In order to better understand
the potential sources of the target substances, we performed correlation analysis between the
measured levels of N/OPAHs and other PM constituents, namely, WSOC, HULIS, nitrate,
sulphate, and $K^+$. We found a significant correlation ($n = 5$, p<0.05) between $9,10-O_2ANT$ and 1,2-
$O_2BAA$ at TK site, which suggests a common emission source (Table S6). The data shown in





Tables S6 and S7 also indicate significant correlations ($p < 0.05$) between the levels of BaOFLN
and 1-NPYR (produced by primary sources), and WSOC, HULIS, and $K^+$ (BB marker) in TK
samples. 1-NPYR is the predominant congener among NPAHs found in diesel engine exhaust
particles and was proposed as marker for diesel emission (Bamford et al., 2003; IARC 2013), but
it may also be emitted with relatively small quantities from biomass-fuelled combustion (Shen et
al., 2012; Orakij et al., 2017). These findings suggest the importance of primary emission sources
including BB and diesel exhaust in TK study area. For MZ samples, we found significant
correlations ($n = 4$, $p < 0.05$) of 9-OFLN, BaOFLN, and 9-NANT with WSOC and HULIS, without
any significant correlations to $K^+$, suggesting the presence of mixed air masses that were fed by
both primary and secondary sources at MZ site. The absence of both NPYR isomers in MZ samples,
which are indicative of road traffic and industrial emissions and long-range transported pollution
(IARC, 1989; Finlayson-Pitts and Pitts, 2000; Lammel et al., 2017), indicates that chemically aged
air was advected during the MZ campaign (Voliotis et al., 2017).

**3.4 Mass size distribution of N/OPAHs**
N/OPAH MSDs are shown in Figs. 1 and 2. On average, the MMDs of NPAHs were 0.06 µm at
MZ and 0.12 µm at TK, while those for OPAHs were 0.06 µm at MZ and 0.10 µm at TK. The
MMDs for quinones were 0.07 and 0.15 at the two sites, respectively. We found two distinct MSD
patterns among the samples: the first pattern observed in three samples across the two sites (one
sample set from MZ and two sets from TK; Figs. S3c, S4a, d and e), was dominated by OBAT
followed by BbOFLN. The MMD of OPAHs in these three samples was on average 0.06 µm
(ranging within 0.05-0.09 µm). The unique analyte distribution in these samples was accompanied
by a noticeably higher enrichment in $PM_{0.49}$ as well as relatively high concentrations compared to
the rest of samples. The preferential enrichment of OBAT, BaOFLN, and BbOFLN in sub-
micrometre PM was previously reported from locations in Europe, Asia, and the USA (Allen et al.,
1997; Albinet et al., 2008; Ladji et al., 2009; Ringuet et al., 2012; Shen et al., 2016; Gao et al.,
2019). The observed pattern could be the evidence of fresh emission from primary sources, as was
discussed in the previous section. The second pattern, which was seen in the remaining six sample
sets, was considerably different: the target substances were more evenly distributed across different
PM size ranges, and often dominated by relatively high abundance of quinones, $9,10-O_2ANT$ and
$1,2-O_2BAA$ – the two quinones were previously reported with preferential enrichment in ultrafine


PM (Ringuet et al., 2012; Shen et al., 2016). The MMD of OPAHs in these five sample sets was
on average 0.25 μm (ranging within 0.08-0.49 μm). This distribution points at relatively aged air
masses and the contribution of both primary and secondary sources.
In terms of the inter-site variability of target substance MSD, the size fraction $PM_{0.49}$ was more
prominent in MZ than in TK, i.e. on average 74% for NPAHs, 75% for OPAHs, 69% for quinones
at MZ, compared to 55, 60, and 52%, respectively, at TK site (Figs. 1-2 and S3-5). The largest
differences found among each substance group were for 9-NANT (28% higher at MZ), BbOFLN
(25% higher), and $1,2-O_2BAA$ (17% higher). The values for NPAHs from TK were lower than
those previously found for wintertime PM at this site (59 and 71% for a traffic and urban
background site, respectively; Besis et al., 2017). The higher enrichment of predominant NPAHs
(9-NANT and 2-NFLT; Figure S3-S4) in $PM_{0.49}$ in the present study is in agreement with the MSDs
reported for these compounds from several other locations in Europe and Asia (Ringuet et al., 2012;
Lan et al., 2014; Lammel et al., 2017). The preferential enrichment of N/OPAHs in sub-micron
PM, especially $PM_{0.49}$, raises concerns with respect to the inhalation toxicity of airborne PM; this
is because $PM_{0.49}$ is capable of reaching deeper regions in the lung. This is exacerbated by the
ability of quinones to catalyse redox reactions and the formation of ROS in the respiratory system.

**3.5 NMAHs and N/OPAHs as part of HULIS**
Because of their water-solubility, NMAHs are constituents of PM HULIS and WSOC (Claeys et
al., 2012; Teich et al., 2017). This substance class contributed ≈0.4 and 1.8% to HULIS mass at
the MZ and TK sites, respectively (Table 3). This contribution was fairly even across the size
fractions addressed, while showing a maximum for particles size 0.95-3 μm, namely ≈0.7 and 2.0%
by mass at the MZ and TK sites, respectively. The large particle size, 0.95-3 μm, points to the
significance of aqueous phase processes and in general slower formation of NMAHs (Voliotis et
al., 2017). The water-soluble N/OPAHs, i.e. $1,4-O_2NAP$ and 1-NNAP, contributed up to 0.0006 %
to HULIS (Table 3; water solubility of ≥50 mg $L^{-1}$ was used as criterion), with values peaking in
the PM size fractions 0.95-3 and >3 μm at MZ and TK, respectively. Similar to NMAHs, the
N/OPAH mass mixing ratios in HULIS did not significantly vary with particle size (Table 3).
Our reported NMAH contribution to HULIS mass is in good agreement with the results of previous
reports from urban sites in Europe (Kitanovski et al., 2012b; Claeys et al., 2012) and Brazil (Caumo
et al., 2016). Specifically, Kitanovski et al. (2012) found that NMAHs contributed 0.4-1.3% to the





winter urban $PM_{10}$ OC mass from Ljubljana (Slovenia), while in another study, 4-NC alone
contributed 0.46% and 0.04% to the HULIS mass in urban spring and summer $PM_{2.5}$ from Budapest
(Hungary), respectively (Claeys et al., 2012). Moreover, NMAHs (4-NP, 4-NC, MNCs and
dimethyl-nitrocatechols (DMNCs)) contributed 0.28% and 0.35% to the OC mass in winter $PM_{10}$
samples from São Paulo, Brazil (Caumo et al., 2016). Lower NMAH contribution to HULIS (or
OC) mass were reported for rural sites in Europe. For example, 4-NC contributed 0.03% to the
HULIS mass in summer $PM_{2.5}$ from K-puszta, Hungary (Claeys et al., 2012), while total NMAHs
(NPs, 4-NC, MNCs and DMNCs) presented 0.75% of OC mass in winter $PM_{10}$ sampled at a rural
background site in Belgium (Kahnt et al., 2013).
In Sect. 3.2 we emphasized the similar MSDs at both locations between HULIS on one side and
NCs and NSAs on the other. These two NMAH subclasses on average contributed to ≈83% and
≈94% of total NMAHs in $PM_{0.95}$, and ≈55% and 87% of total NMAHs in $PM_{3-0.95}$ at MZ and TK
sites, respectively (Table S8). At both sites, NCs were the dominant NMAH species. It is also
interesting to note that HULIS showed higher correlations with NSAs and NCs in MZ ($R^2_{adj}$ 0.68-
0.98; Table S5), than in TK ($R^2_{adj}$ 0.24-0.59; Table S4). BaOFLN and 1-NPYR, as well as 9-OFLN,
BaOFLN and 9-NANT showed similar MSDs and significant correlations ($R^2_{adj} \geq 0.8$; Tables S6-
S7) with HULIS at TK and MZ, respectively, suggesting that these N/OPAHs are most likely
constituents of the HULIS. The significant correlations in the levels of 1-NPYR with HULIS,
WSOC, and $K^+$ (Table S6) are particularly interesting, as 1-NPYR is exclusively associated with
primary emission sources. These observations are in line with our previous discussion that MZ site
was mainly influenced by aged air masses, while TK site by a mixture of fresh (BB and fossil fuel)
emissions and aged aerosols (Voliotis et al., 2017).
With mass mixing ratios of the order of 1%, NMAHs are constituents of HULIS with limited
significance by mass, but their relevance is more significant due to their optical properties (Mohr
et al., 2013; Laskin et al., 2015; Teich et al., 2017; Xie et al., 2017). Teich et al. (2017) found that
the mass contributions of total NMAHs (NPs and NSAs) to WSOC on average was five times lower
than their contribution to the light absorption of the aqueous PM extract at 370 nm (Teich et al.,
2017). This implies that even small fractions of chromophoric HULIS compounds such as NMAHs
and water soluble N/OPAHs can have an excessive influence on the aerosol light absorption (Mohr
et al., 2013; Teich et al., 2017) and the atmospheric photochemical processes, especially in polluted
areas (Laskin et al., 2015; Teich et al., 2017).





**4. Final remarks**
We studied the composition and MSDs of NMAHs and N/OPAHs in PM from urban locations in
Germany and Greece, with some of the target substances (i.e. NSAs, MNCs and MNPs) studied in
size-resolved PM for the first time. At both locations, NCs were the most abundant NMAH species,
and OPAHs were more abundant and more frequently detected than NPAHs. The total
concentrations of the most abundant NMAHs, NCs, and N/OPAHs were up to 10 times higher in
TK than in MZ. Correlation analysis of NMAHs revealed distinct features among the sites,
suggesting mixed air masses influenced by fresh BB and aged fossil fuel combustion emissions at
TK, and aged advected air influenced by combustion emissions (i.e. BB) at MZ.
The MSDs of NMAHs, OPAHs and NPAHs were rather similar, but exhibited temporal and spatial
variations due to daily changes in atmospheric conditions and different sources. In general, NCs,
NSAs, OPAHs and NPAHs showed unimodal MSDs peaking in the finest PM fraction, $PM_{0.49}$,
which was more prominent in MZ than in TK. NPs exhibited bimodal MSDs with the dominant
peak in $PM_{0.49}$. The MMDs of all chemical classes were lower at MZ than at TK. Larger MMDs at
TK could be explained by the larger PM size range collected at this site, but they could also be an
indication of aerosol aging. On average, NMAHs and water-soluble N/OPAHs (i.e. $1,4-O_2NAP$
and 1-NNAP) contributed up to 1.8 and 0.0006% to the HULIS mass in the study areas. Although
NMAHs and N/OPAHs represent a small fraction of PM HULIS (and WSOC), due to their light
absorption properties, their impact on the total aerosol light absorption could be disproportionally
large. This is particularly important for atmospheric photochemical processes in polluted areas.

**Acknowledgements**
We thank Eleni Papakosta (Prefecture of Thessaloniki), Thorsten Hoffmann and Anna Honcza
(Max Planck Institute for Chemistry) for onsite and laboratory support. This research was
supported by the Max Planck Society and the Postgraduate Program "Environmental Chemistry
and Pollution Control" of the Aristotle University of Thessaloniki.

**Author contributions.** GL and CS conceived the study. PS and AV conducted the air sampling
and field measurements. ZK and PS did the chemical analysis of samples. ZK, PS, and GL did the
data analysis. ZK, PS, and GL discussed the results and wrote the manuscript with input from all
co-authors.



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

Contributions of nitrated aromatic compounds to the light absorption of water-soluble and particulate brown carbon in
different atmospheric environments in Germany and China, Atmos. Chem. Phys., 17, 1653-1672, doi:10.5194/acp-17-
756 1653-2017, 2017.

Tomaz, S., Shahpoury, P., Jaffrezo, J.-L., Lammel, G., Perraudin, E., Villenave, E. and Albinet, A.: One-year study of
polycyclic aromatic compounds at an urban site in Grenoble (France): seasonal variations, gas/particle partitioning and
cancer risk estimation, Sci. Total Environ., 565, 1071–1083, doi:10.1016/j.scitotenv.2016.05.137, 2016.
van Pinxteren, D. and Herrmann, H.: Determination of functionalised carboxylic acids in atmospheric particles and
cloud water using capillary electrophoresis/mass spectrometry, J. Chromatogr. A, 1171, 112-123,
doi:10.1016/j.chroma.2007.09.021, 2007.
van Pinxteren, D., Teich, M., and Herrmann, H.: Hollow fibre liquid-phase microextraction of functionalised
carboxylic acids from atmospheric particles combined with capillary electrophoresis/mass spectrometric analysis, J.
Chromatogr. A, 1267, 178-188, doi:10.1016/j.chroma.2012.06.097, 2012
Verma, V., Fang, T., Guo, H., King, L., Bates, J. T., Peltier, R. E., Edgerton, E., Russell, A. G., and Weber, R. J.:
Reactive oxygen species associated with water-soluble PM2.5 in the southeastern United States: spatiotemporal trends
and source apportionment, Atmos. Chem. Phys., 14, 12915-12930, doi:10.5194/acp-14-12915-2014, 2014.



Verma, V., Wang, Y., el Afifi, R., Fang, T., Rowland, J., Russell, A.G. and Weber, R.J.: Fractionating ambient humic-
like substances (HULIS) for their reactive oxygen species activity - assessing the importance of quinones and
atmospheric aging. Atmos. Environ., 120, 351-359, doi:10.1016/j.atmosenv.2015.09.010, 2015.
Vicente, E. D., Vicente, A. M., Musa Bandowe, B. A. and Alves, C. A.: Particulate phase emission of parent polycyclic
aromatic hydrocarbons (PAHs) and their derivatives (alkyl-PAHs, oxygenated-PAHs, azaarenes and nitrated PAHs)
from manually and automatically fired combustion appliances, Air Qual. Atmos. Heal., 9, 653–668,
doi:10.1007/s11869-015-0364-1, 2016.
Vione, D., Maurino, V., and Minero, C.: Photosensitized humic-like substances (HULIS) formation processes of
atmospheric significance: a review, Environ. Sci. Pollut. Res. 21, 11614-11622, doi:10.1007/s11356-013-2319-0,
778 2014.

Voliotis A., Prokeš R., Lammel G., and Samara C.: New insights on humic-like substances associated with urban
aerosols from central and southern Europe: size-resolved chemical characterization and optical properties. Atmos.
Environ., 166, 286-299, doi:10.1016/j.atmosenv.2017.07.024, 2017.
Walgraeve, C., Demeestere, K., Dewulf, J., Zimmermann, R. and van Langenhove, H.: Oxygenated polycyclic
aromatic hydrocarbons in atmospheric particulate matter: Molecular characterization and occurrence, Atmos. Environ.,
44, 1831–1846, doi:10.1016/j.atmosenv.2009.12.004, 2010.
Wang, L., Wang, X., Gu, R., Wang, H., Yao, L., Wen, L., Zhu, F., Wang, W., Xue, L., Yang, L., Lu, K., Chen, J.,
Wang, T., Zhang, Y., and Wang, W.: Observations of fine particulate nitrated phenols in four sites in northern China:
concentrations, source apportionment, and secondary formation, Atmos. Chem. Phys., 18, 4349-4359,
doi:10.5194/acp-18-4349-2018, 2018.
Wang, X., Gu, R., Wang, L., Xu, W., Zhang, Y., Chen, B., Li, W., Xue, L., Chen, J., and Wang, W.: Emissions of fine
particulate nitrate phenols from the burning of five common types of biomass, Environ. Pollut., 230, 405-412,
doi:10.1016/j.envpol.2017.06.072, 2017.
Wang, Y., Hu, M., Wang, Y., Zheng, J., Shang, D., Yang, Y., Liu, Y., Li, X., Tang, R., Zhu, W., Du, Z., Wu, Y., Guo,
S., Wu, Z., Lou, S., Hallquist, M., and Yu, J.: The formation of nitro-aromatic compounds under high NOx-
anthropogenic VOCs dominated atmosphere in summer in Beijing, China, Atmos. Chem. Phys., 19, 7649-7665,
doi:10.5194/acp-19-7649-2019, 2019.
Xie, M.J., Chen, X., Hays, M.D., Lewandowski, M., Offenberg, J., Kleindienst, T.E. and Holder, A.L.: Light
absorption of secondary organic aerosol: composition and contribution of nitroaromatic compounds. Environ. Sci.
Technol., 51, 11607–11616, doi:10.1021/acs.est.7b03263, 2017.
Zhang, X., Hecobian, A., Zheng, M., Frank, N.H. and Weber, R.J.: Biomass burning impact on PM $_{2.5}$ over the
southeastern US during 2007: integrating chemically speciated FRM filter measurements, MODIS fire counts and PMF
analysis, Atmos. Chem. Phys., 10, 6839–6853, doi:10.5194/acp-10-6839-2010, 2010.
Zheng, G., He, K., Duan, F., Cheng, Y. and Ma, Y.: Measurement of humic-like substances in aerosols: a review,
Environ. Pollut., 181, 301–314, doi:10.1016/j.envpol.2013.05.055, 2013.
Zhuo, S., Du, W., Shen, G., Li, B., Liu, J., Cheng, H., Xing, B. and Tao, S.: Estimating relative contributions of primary
and secondary sources of ambient nitrated and oxygenated polycyclic aromatic hydrocarbons, Atmos. Environ., 159,
126–134, doi:10.1016/j.atmosenv.2017.04.003, 2017.
Zielinska, B., Sagebiel, J., Arnott, W.P., Rogers, C.F., Kelly, K.E., Wagner, D.A., Lighty, J.S., Sarofim, A.F., and
Palmer, G.: Phase and size distribution of polycyclic aromatic hydrocarbons in diesel and gasoline vehicle emissions,
Environ. Sci. Technol., 38, 2557-2567, doi:10.1021/es030518d, 2004.



**Table 1.** Analytes targeted in this study

| Analyte | Abbreviation | Q1 |
|---|---|---|
| 3-Nitrosalicylic acid | 3-NSA | 182 |
| 5-Nitrosalicylic acid | 5-NSA | 182 |
| 4-Nitrocatechol | 4-NC | 154 |
| 4-Nitroguaiacol | 4-NG | 168 |
| 4-Methyl-5-nitrocatechol | 4-M-5-NC | 168 |
| 4-Nitrophenol | 4-NP | 138 |
| 2,4-Dinitrophenol | 2,4-DNP | 183 |
| 3-Methyl-4-nitrophenol | 3-M-4-NP | 152 |
| 3-Methyl-5-nitrocatechol | 3-M-5-NC | 168 |
| 3-Methyl-4-nitrocatechol | 3-M-4-NC | 168 |
| 2-Methyl-4-nitrophenol | 2-M-4-NP | 152 |
| 2-Methyl-3,5-dinitrophenol (Dinitro-ortho-cresol) | DNOC | 197 |
| 1-Nitronaphthalene | 1-NNAP | 173.1 |
| 2-Nitronaphthalene | 2-NNAP | 173.1 |
| 5-Nitroacenaphthene | 5-NACE | 199.1 |
| 2-Nitrofluorene | 2-NFLN | 211.1 |
| 9-Nitroanthracene | 9-NANT | 223.1 |
| 9-Nitrophenanthrene | 9-NPHE | 223.1 |
| 3-Nitrophenanthrene | 3-NPHE | 223.1 |
| 2-Nitrofluoranthene | 2-NFLT | 247.1 |
| 3-Nitrofluoranthene | 3-NFLT | 247.1 |
| 1-Nitropyrene | 1-NPYR | 247.1 |
| 2-Nitropyrene | 2-NPYR | 247.1 |
| 7-Nitrobenz(a)anthracene | 7-NBAA | 273.1 |
| 6-Nitrochrysene | 6-NCHR | 273.1 |
| 1,3-Dinitropyrene | 1,3-N$_2$PYR | 292.1 |
| 1,6-Dinitropyrene | 1,6-N$_2$PYR | 292.1 |
| 1,8-Dinitropyrene | 1,8-N$_2$PYR | 292.1 |
| 6-Nitrobenz(a)pyrene | 6-NBAP | 297.1 |
| 1,4-Naphthoquinone | 1,4-O$_2$NAP | 158.1 |
| 9-Fluorenone | 9-OFLN | 180.1 |
| 9,10-Anthraquinone | 9,10-O$_2$ANT | 208.1 |
| 2-Nitro-9-fluorenone | 2-N-9-OFLN | 225.1 |
| Benz(a)fluorenone | BaOFLN | 230.1 |
| Benz(b)fluorenone | BbOFLN | 230.1 |
| Benzanthrone | OBAT | 230.1 |
| 1,2-Benzanthraquinone | 1,2-O$_2$BAA | 258.1 |

Q1 – $m/z$ of ions used for quantification in ESI(−)MS for NMAHs and NCI-MS for N/OPAHs



**Table 2.** Sampling details

| | Cut-off diameters (µm) | Sampling date | Sample volume (m³) |
|---|---|---|---|
| **Mainz** [a] **49.99° N 8.23° E** | 10 - 7.2 <br> 7.2 - 3 <br> 3 - 1.5 <br> 1.5 - 0.95 <br> 0.95 - 0.49 <br> <0.49 | 17.-20.11.2015 <br> 26.-29.11.2015 <br> 01.-04.12.2015 <br> 04.-07.12.2015 | 3402 <br> 4124 <br> 4088 <br> 4197 |
| **Thessaloniki 40.63°N 22.96° E** | 10 – 3 [b] <br> 3 - 0.95 [b] <br> 0.95 - 0.49 <br> <0.49 | 27.-29.1.2016 <br> 08.-10.2.2016 <br> 16.-18.2.2016 <br> 22.-24.2.2016 <br> 17.-19.3.2016 | 3228 <br> 3228 <br> 3228 <br> 3172 <br> 3175 |

[b] pooled from two impactor stages





**Table 3**. Mean absolute concentrations and mass mixing ratios (in brackets) of HULIS[a] in WSOC[a] as well as of NMAHs and water-soluble (WS) N/OPAHs[b] in HULIS in (a) Mainz and (b) Thessaloniki PM.

**a**.

| Particle size μm | WSOC (μgC m$^{-3}$) | HULIS μg m$^{-3}$ (% C/C) | NMAHs ng m$^{-3}$ (%) | WS N/OPAHs pg m$^{-3}$ (%) |
|---|---|---|---|---|
| < 0.49 | 1.14 | 0.80 (39) | 3.41 (0.43) | 0.7 (0.0001) |
| 0.49-0.95 | 0.68 | 0.31 (25) | 1.24 (0.40) | 0.2 (0.0001) |
| 0.95-3 | 0.18 | 0.09 (28) | 0.65 (0.73) | 0.3 (0.0003) |
| 3-10 | 0.12 | 0.09 (42) | 0.27 (0.30) | 0.2 (0.0002) |
| Total | 2.07 | 1.29 (33) | 5.58 (0.43) | 1.4 (0.0001) |

**b**.

| Particle size (μm) | WSOC (μgC m$^{-3}$) | HULIS μg m$^{-3}$ (% C/C) | NMAHs ng m$^{-3}$ (%) | WS N/OPAHs pg m$^{-3}$ (%) |
|---|---|---|---|---|
| < 0.49 | 2.02 | 1.29 (34) | 24.0 (1.9) | 2.7 (0.0002) |
| 0.49-0.95 | 1.28 | 0.83 (34) | 13.9 (1.7) | 0.7 (0.0001) |
| 0.95-3 | 0.57 | 0.35 (32) | 6.89 (2.0) | 0.5 (0.0001) |
| > 3 | 0.33 | 0.11 (18) | 1.87 (1.7) | 0.7 (0.0006) |
| Total | 4.20 | 2.58 (32) | 46.6 (1.8) | 4.5 (0.0002) |

[a] Voliotis et al., 2017
[b] 1,4-$O_2$NAPs and 1-NNAP (criteria: water solubility > 50 mg L$^{-1}$)



**Figure 1**. Mass size distributions (MSDs) of PM-bound NMAHs, N/OPAHs, WSOC, HULIS and ions in Mainz (Germany). The error bars represent standard deviations. [a] compound MSD calculated from one (out of four) sample set (detected and quantified in one sample set only); [b] compound MSD calculated from three (out of four) sample sets (detected and quantified in three sample sets only)



**Figure 2.** Mass size distributions (MSDs) of PM-bound NMAHs, N/OPAHs, WSOC, HULIS and ions in Thessaloniki (Greece). The error bars represent standard deviations. [b] compound MSD calculated from three (out of five) sample sets (detected and quantified in three sample sets only)