# Peer review of "Composition and mass size distribution of nitrated and oxygenated aromatic compounds in ambient particulate matter from southern and central Europe – implications for origin"

_Atmospheric Chemistry and Physics, 2019_

## Referee Comment (RC1) · Anonymous Referee #1 · 11 Sep 2019

This study quantified some important nitro-aromatic species and nitrated/oxygenated-PAHs at two urban sites in Europe. The mass size distributions and sources of targeted species are analyzed. The topic is valuable, while the manuscript is not so well organized and sometimes is hard to follow. Some important conclusions drawn in this study are not very validated and sometimes inconsistent, especially for the source analysis (pls see following major comments), which is important part of the paper. The following major comments need to be carefully addressed before re-considered by ACP.

1. The discussion on the sources or origins of NMAHs and O/NPAHs is important

part of the manuscript, while related discussions in the main text cannot support the conclusions and sometimes inconsistent. The authors concluded that fresh biomass burning emissions dominated at the TK site, while aged air masses were predominant at the MZ site in the abstract (lines 18-20), however:

Lines 263-293: The authors hypothesize the TK site was greatly influenced by BB emissions based on the high concentrations of NC, MNC and NSA. This is also an important conclusion drawn in the abstract (lines 18-19: fresh biomass burning emissions dominated at the TK site). However, the authors found NC and MNC showed low correlations with K+ (biomass burning tracer) in line 274. And then draw the conclusion that gas-phase nitration of anthropogenic precursors, rather than BB, was the main source of NC and MNC in lines 283-286. NC and MNCs are the most abundant species among the quantified NMAHs in this study, so the bad correlations with K+ indicated the less importance of BB emissions at TK site.

If the conclusion in line 266 (The air masses over TK during sampling were greatly influenced by BB emissions) is true, why not use the fire maps and back trajectories during the sampling to validate this?

Again, the authors concluded that aged air masses were predominant at MZ site in lines 19-20. However, good correlations between K+ and NC, MNCs were reported in lines 296-297. NC and MNC are the most abundant NMAHs at MZ site (lines 246-247). Thus, BB emission seems the more important source of NMAHs at the MZ site.

Lines 297-303: The formation of NSA, NP, MNP and DNP via aqueous-phase nitration may be arbitrary. For example, 4-NP can form via gas-phase oxidation and then partition into the particle phase. There is another possibility that the nitrate (very hygroscopic) facilitates the aerosol absorbing more water, resulting in higher aerosol liquid water content, which promote the gas-phase NPs or other NMAHs partitioning into the particles. In this case, nitrate and the NMAHs can also show good correlations.

Lines 418-421: How could the correlations between the analyzed species and WSOC

or HULIS, and bad correlations between analyzed species and K+ suggest the sources from both primary and secondary sources? Bad correlations with K+ indicate the less contribution of BB emissions. Both WSOC and HULIS have primary BB emission and secondary formation sources.

What's more, there were only four or five points to do the correlations, which I think it's not quite convincing to analyze the sources.

2. Section 3.5 lines 461-470 and lines 11-13: This study calculated the contributions of NMAHs to HULIS and WSOC. HULIS and WSOC are just the water soluble organics, while NMAHs were extracted by methanol in this study. I agree that NMAHs could be the constituents among HULIS and WSOC, while using methanol-extracted NMAHs to calculate their contributions to HULIS or WSOC may not reasonable. I am not sure if the methanol extracted fractions are equal to the water soluble fractions in WSOC or HULIS (I think the NMAHs among HULIS or WSOC may be lower than the methanol-extracted NMAHs). Furthermore, some of the water-extracted NMAHs could be excluded from the HULIS fraction during the SPE processes. Can you show data (e.g. the ratios of water-extracted/methanol-extracted NMAHs) or any evidence to suggest the calculation is reasonable? Also, N/O-PAHs are extracted by dichloromethane. Are the water-extracted N/O-PAHs in HULIS or WSOC are equal to the dichloromethane-extracted fractions?

Lines 331-333: I think similar mass size distributions of NMAHs and HULIS only indicate they may have similar atmospheric processes, and cannot indicate NC and NSAs are important constituents of HULIS.

3. Lines 260-261 and related analysis throughout the paper: How do the authors know the different concentration levels of the two sites are not due to different cut-off of the aerosol samples? I notice that both sites have the same cut-off of 0.49-0.95, 0.95-1.5, 1.5-3, 3-7.2 $\mu$m, it would be meaningful to compare the concentrations of the same particle size ranges and then analyze the different sources at the two sites.

4. Some important information should be added in the Experimental section:

1) Section 2.2: Why sampling is conducted at the two sites? They are both urban sites. Please specify the differences of the two sites in section 2.2, such as referring previous studies in the two sites to address the differences. It is not quite clear for me what's the differences between the two sites even after finish reading the paper.

2) Section 2.2 and Table 2: Only four sets of sample were collected at MZ site, and five sets at TK site. How can the authors confirm that only four or five samples can represent the conditions at the sites?

3) Section 2.3: Please show the recoveries of the quantified species in Table 1. And pls show the compound peaks and resolution of NMAHs in the LC-MS chromatograms.

4) The extraction and quantification of K+, WSOC and HULIS are omitted in the experimental section.

5. Besides, I suggest the authors to carefully check the data and analysis throughout the paper, and to draw scientific and validated conclusions. A better proofing reading is also needed.

Specific comments:

Lines 71-73: This sentence is confusing. Which source is more dominated in urban areas? Primary sources or secondary formation? Pls revise to be clear.

Lines 103-105: I think these three references are more focused on the light absorption of nitro-aromatics. Could the authors add more references suggesting the light absorption capacity of PAHs and N/O-PAHs?

Lines 309-401: Please clarify what are "these OPAHs" and what are "other N/OPAHs".

Lines 465-466: Could the authors further explain why larger particle size suggests aqueous-phase processes and slower formation?

---

## Referee Comment (RC2) · Anonymous Referee #2 · 2 Oct 2019

The paper reports on findings from wintertime measurements of composition and mass size distribution of different nitrated and oxygenated aromatics in Mainz, Germany and Thessaloniki, Greece. Correlation coefficients between the concentration of these species and WSOC, HULIS, K+, and nitrate were determined to investigate sources of the observed N/O aromatics. The authors conclude that air masses sampled in Thessaloniki were impacted by fresh biomass burning while aged air masses (biomass and fossil fuel combustion) were sampled in Mainz. There is a lot of information on the total concentration of the various species and their size-dependent concentration, so

there's certainly value to having this information for these two cities (despite the short duration of the measurements). However the way the paper is structured and the use of this many acronyms make the paper very hard to read. The other major comment I have is about the conclusions of the source attributions. For example nitrate aerosols could be high in biomass burning plumes as well as aged urban plumes, so I'm not sure a correlation can be really conclusive. Another support for the source apportionment conclusions is the mass size distributions; however the resolution of these distributions is so low that I don't think they can be robust for such interpretation. The other comments are highlighted below. I recommend major revisions and reconsideration before accepting the paper for publication.

Line 9: what does the index in the summation sign indicate? It's probably the number of NMAHs, but perhaps it's more clear if it's defined for at least one group of compounds first. Line 151: what's the time resolution of the samples in TK? I believe it flows better if section 2.2 is presented in the beginning of Section 2, followed by sample preparation and analytical methods. I also think section 2.1 (Chemicals and Solutions) can be moved to SI. L185: each filter paper or just sections of it? L230: what justifies assuming that measurements at TK were also PM10? The authors later on do comment that perhaps larger than 10 um particles were sampled in TK (the statement on L 328-329). L255-256: name of the country (Slovania and China, etc) shouldn't be in () with the reference. L320: how can the contribution from primary traffic emissions explain the peak in MSD in the 0.95-1.5 um range? Primary emissions are typically peaking in <100 nm in number distribution, which puts the mass distribution peak at much smaller than 0.95-1.5 um. L422: This sentence doesn't make sense. I thought NPYR is a marker for primary combustion; so why is "long range transported pollution" also included here? Despite this, the authors claim that lack of NPYR isomers suggests advection of chemically aged plumes to MZ. Aren't these sentences contradictory?
* * *

---

## Author Comment (AC1) · 22 Dec 2019

The comment was uploaded in the form of a supplement:
https://www.atmos-chem-phys-discuss.net/acp-2019-673/acp-2019-673-AC1-supplement.pdf
* * *

---

## Author Response (AR1)

**Authors' responses to Reviewers' comments**

**Manuscript ID:** acp-2019-673

**Title:** Composition and mass size distribution of nitrated and oxygenated aromatic compounds in ambient particulate matter from southern and central Europe - implications for origin

**Authors:** Zoran Kitanovski, Pourya Shahpoury, Constantini Samara, Aristeidis Voliotis, Gerhard Lammel
* * *
**Anonymous Referee #1**

1) Reviewer's general comment:

This study quantified some important nitro-aromatic species and nitrated/oxygenated- PAHs at two urban sites in Europe. The mass size distributions and sources of targeted species are analyzed. The topic is valuable, while the manuscript is not so well organized and sometimes is hard to follow. Some important conclusions drawn in this study are not very validated and sometimes inconsistent, especially for the source analysis (pls see following major comments), which is important part of the paper. The following major comments need to be carefully addressed before re-considered by ACP.

*Authors' response:*

We thank the Reviewer for her/his valuable comments. We have carefully addressed these comments, as explained below.

2) Reviewer's comment:

The discussion on the sources or origins of NMAHs and O/NPAHs is important part of the manuscript, while related discussions in the main text cannot support the conclusions and sometimes inconsistent. The authors concluded that fresh biomass burning emissions dominated at the TK site, while aged air masses were predominant at the MZ site in the abstract (lines 18-20), however:

Lines 263-293: The authors hypothesize the TK site was greatly influenced by BB emissions based on the high concentrations of NC, MNC and NSA. This is also an important conclusion drawn in the abstract (lines 18-19: fresh biomass burning emissions dominated at the TK site). However, the authors found NC and MNC showed low correlations with K+ (biomass burning tracer) in line 274. And then draw the conclusion that gas-phase nitration of anthropogenic precursors, rather than BB, was the main source of NC and MNC in lines 283-286. NC and MNCs are the most abundant species among the quantified NMAHs in this study, so the bad correlations with K+ indicated the less importance of BB emissions at TK site.

*Authors' response:*

At Thessaloniki (TK), the low correlations between the most abundant species, 4-NC and MNCs with $K^+$, WSOC and HULIS indeed indicated relatively low importance of biomass burning in the formation of 'NCs'. However, the significant correlation of WSOC and HULIS with $K^+$ at TK suggests the influence of fresh biomass burning emissions on the PM organic content ($K^+$ is a primary biomass burning marker). Therefore, we conclude that at TK, an interplay of different sources, dominated by fresh emissions, was evident for the time of sampling. This conclusion aligns well with our understanding of N/OPAH sources at TK.

We made the appropriate changes in the abstract and the current Section 3.1.1.

**Note. we have re-structured the manuscript in response to a comment from Reviewer 2.**

Page 1, Lines 16-18: "*Correlation analysis between NMAHs, NPAHs and OPAHs from one side and WSOC, HULIS, sulphate and potassium from another, suggested that fresh biomass burning and fossil fuel combustion emissions dominated at the TK site, while aged air masses were predominant at the MZ site.*"

Page 9, Line 272-274: "*In conclusion, the emission profile and correlation analysis for NMAHs at TK suggest a complex interplay of different emission sources, particularly dominated by fresh BB and fossil fuel combustion emissions.*"

3) Reviewer's comment:

If the conclusion in line 266 (The air masses over TK during sampling were greatly influenced by BB emissions) is true, why not use the fire maps and back trajectories during the sampling to validate this?

*Authors' response:*

Biomass burning in the Thessaloniki area is dominated by household heating within the city and immediate surroundings, i.e. local emissions (Saffari et al. 2013; Velali et al., 2019), that are not detectable by remote sensing (Kaiser et al., 2012) and, for the same reason, performing air mass trajectory analysis cannot be justified in our study.

*Kaiser, J.W., Heil, A., Andreae, M.O., Benedetti, A., Chubarova, N., Jones, L., Morcrette, J.J., Razinger, M., Schultz, M.G., Suttie, M. and van der Werf, G.R.: Biomass burning emissions estimated with a global fire assimilation system based on observed fire radiative power, Biogeosci., 9, 527-554, doi:10.5194/bg-9-527-2012, 2012.*

*Saffari, A., Daher, N., Samara, C., Voutsa, D., Kouras, A., Manoli, E., Karagkiozidou, O., Vlachokostas, C., Moussiopoulos, N., Shafer, M.M., Schauer, J.J., Sioutas, C.: Increased Biomass Burning Due to the Economic Crisis in Greece and Its Adverse Impact on Wintertime Air Quality in Thessaloniki, Environ Sci Technol., 47, 13313-13320, doi:10.1021/es403847h, 2013.*

*Velali, E., Pantazaki, A., Besis, A., Choli-Papadopoulou, T., Samara, C.: Oxidative stress, DNA damage, and mutagenicity induced by the extractable organic matter of airborne particulates on bacterial models, Regul. Toxicol. Pharmacol., 104, 59-73, doi:10.1016/j.yrtph.2019.03.004, 2019.*

4) Reviewer's comment:

Again, the authors concluded that aged air masses were predominant at MZ site in lines 19-20. However, good correlations between K+ and NC, MNCs were reported in lines 296-297. NC and MNC are the most abundant NMAHs at MZ site (lines 246-247). Thus, BB emission seems the more important source of NMAHs at the MZ site.

*Authors' response:*

We agree that biomass burning is a significant source of NMAHs at MZ due to the good correlations of 4-NC and MNCs with $K^+$. Therefore, we edited the text throughout the Section 3.1.1 of the revised manuscript and added the following conclusion regarding MZ site:

Page 9, Line 285- Page 10, Line 290: "*The significant correlations of 4-NC and MNCs with 4-NP and MNPs in $PM_{0.95}$ and $PM_{3-0.95}$, suggest similar sources for NCs and NPs over MZ. Moreover, high correlations of 4-NC and MNC with $K^+$ in $PM_{0.95}$ indicate that BB was a significant emission source over MZ (Chow et al., 2016; Voliotis et al., 2017; Wang et al., 2018), whereas their high correlations with sulphate in $PM_{3-0.95}$ ($0.66 < R^2_{adj} < 1.00$; Table S9) could infer possible anthropogenic emissions, i.e. coal combustion (Lu et al., 2019a).*"

5) Reviewer's comment:
Lines 297-303: The formation of NSA, NP, MNP and DNP via aqueous-phase nitration may be arbitrary. For example, 4-NP can form via gas-phase oxidation and then partition into the particle phase. There is another possibility that the nitrate (very hygroscopic) facilitates the aerosol absorbing more water, resulting in higher aerosol liquid water content, which promote the gas-phase NPs or other NMAHs partitioning into the particles. In this case, nitrate and the NMAHs can also show good correlations.

*Authors' response:*
We agree with the comment and removed discussions related to emission sources in relation to nitrate concentrations.

Page 9, Line 281-285: "In the same PM size range, nitrate showed moderate-to-high correlations ($0.5 \leq R^2_{adj} < 0.9$) with NSAs, 4-NP, MNPs, 4-NC and MNCs (Table S8), which are much higher than the corresponding ones in $PM_{0.97}$ samples from TK (Table S5). In $PM_{3-0.95}$, HULIS showed significant correlations with $K^+$ ($R^2_{adj} = 0.99$) and all NMAH species ($0.66 < R^2_{adj} < 0.98$; Table S9), except for 2,4-DNP, suggesting that NMAHs and PM HULIS had similar sources (i.e. BB)."

6) Reviewer's comment:
Lines 418-421: How could the correlations between the analysed species and WSOC or HULIS, and bad correlations between analysed species and K+ suggest the sources from both primary and secondary sources? Bad correlations with K+ indicate the less contribution of BB emissions. Both WSOC and HULIS have primary BB emission and secondary formation sources.

*Authors' response:*
We have revised the related statement in order to reflect the Reviewer's suggestion:

Page 11, Line 349 – Page 12, Line 358: "*For MZ $PM_{10}$ samples, we found significant correlations (n = 4, p<0.05) of 9-OFLN, BaOFLN, and 9-NANT with WSOC and HULIS (Table S13), without any significant correlations to $K^+$. We found similar correlations in $PM_{0.97}$, which suggest the predominance of chemically aged air masses that were advected during the MZ campaign. This is further supported by the absence of NPYR isomers in MZ samples, which are indicative of road traffic and industrial emissions (IARC, 1989; Finlayson-Pitts and Pitts, 2000; Lammel et al., 2017; Voliotis et al., 2017). Finally, $K^+$, WSOC, and HULIS correlated significantly at TK (p<0.05, $R^2_{adj}$ 0.89-0.90), whereas such correlations were not found at MZ. In summary, while N/OPAHs from TK samples were influenced by primary emissions related to BB and fossil fuel combustion, those from MZ samples were dominated by aged air masses.*"

7) Reviewer's comment:

What's more, there were only four or five points to do the correlations, which I think it's not quite convincing to analyse the sources.

*Authors' response:*

Although performed on limited number of samples, we believe that there are interesting patterns in our dataset that deserve to be noted and should not be overlooked. In addressing the Reviewer's concern and for clarification, we have added the following statement to the text:

Page 8, Line 239-241: "*Although the correlation analysis was performed using limited number of sample sets per location (five for TK and four for MZ), it showed several interesting features. Based on these results, we propose the most probable sources for NMAHs at both sampling sites.*"

8) Reviewer's comment:

Section 3.5 lines 461-470 and lines 11-13: This study calculated the contributions of NMAHs to HULIS and WSOC. HULIS and WSOC are just the water soluble organics, while NMAHs were extracted by methanol in this study. I agree that NMAHs could be the constituents among HULIS and WSOC, while using methanol-extracted NMAHs to calculate their contributions to HULIS or WSOC may not reasonable. I am not sure if the methanol extracted fractions are equal to the water soluble fractions in WSOC or HULIS (I think the NMAHs among HULIS or WSOC may be lower than the methanol- extracted NMAHs). Furthermore, some of the water-extracted NMAHs could be excluded from the HULIS fraction during the SPE processes. Can you show data (e.g. the ratios of water-extracted/methanol-extracted NMAHs) or any evidence to suggest the calculation is reasonable? Also, N/O-PAHs are extracted by dichloromethane. Are the water-extracted N/O-PAHs in HULIS or WSOC are equal to the dichloromethane- extracted fractions?

*Authors' response:*

We thank the Reviewer for raising this question and we understand their concerns regarding the water solubility of the studied NMAHs. The NMAHs are indeed water soluble: the water solubility (g/L) of some NMAH species are as follows: 3-nitrosalicylic acid: 1.3 g/L and 5-nitrosalicylic acid: 2.0 g/L (Myrdal et al., 1992); 4-nitrocatechol: 10.5 g/L (estimate EPISuite, USEPA, 2012); 4-methyl-5-nitrocatechol: 4.85 g/L (estimate EPISuite, USEPA, 2012); 4-nitrophenol: 11.6 g/L (Schwarzenbach et al., 1988); 3-methyl-4-nitrophenol: 1.19 g/L (Schwarzenbach et al., 1988); 2,4-dinitrophenol: 2.79 g/L (Schwarzenbach et al., 1988). Our previous paper on NMAHs presented clear evidence of the occurrence of NMAHs in aqueous media (i.e. rainwater and snow; Shahpoury et al., 2018). In that paper, we extracted the NMAHs on SPE disks based on divinylbenzene (DVB) polymer as sorbent with high recoveries (Shahpoury et al., 2018). HULIS is produced similarly by extracting the PM organic material with Oasis HLB sorbent (Voliotis et al., 2017) which is also based on divinylbenzene chemistry. For our manuscript in preparation (Kitanovski et al., 2020), we have determined the polymeric(DVB) SPE recoveries of NMAHs from aqueous artificial lung fluids, i.e. Gamble solution (pH 7.4; Marques et al., 2011) and artificial lysosomal fluid (ALF, pH 4.5; Colombo et al., 2008), which both represent dilute aqueous solutions of inorganic and organic salts. The NMAH recoveries were 95-104% for Gamble solution and 95-105% for ALF, which are similar to recoveries for methanol extracted NMAHs (88-95%). In conclusion, based on literature data and our laboratory results, we are confident that the NMAHs studied in the current work are completely extracted with the rest of the

HULIS material during the HULIS extraction procedure. Therefore, the comparison of the methanol extracted NMAH concentrations with water extracted HULIS concentrations for the same PM samples is, in this case, reasonable. However, we cannot say the same for N/OPAHs which were extracted using dichloromethane. Therefore, we omitted N/OPAHs from the discussion on contribution to HULIS (currently 'Section 3.3 NMAHs as part of HULIS').

Colombo, C., Monhemius, A.J. and Plant, J.A.: Platinum, palladium and rhodium release from vehicle exhaust catalysts and road dust exposed to simulated lung fluids, Ecotox. Environ. Safety, 71, 722-730, doi:10.1016/j.ecoenv.2007.11.011, 2008.

Finewax, Z., de Gouw, J.A. and Ziemann, P.J.: Identification and quantification of 4-nitrocatechol formed from OH and NO$_3$ radical-initiated reactions of catechol in air in the presence of NO$_x$: implications for secondary organic aerosol formation from biomass burning. Environ. Sci. Technol., 52, 1981-1988, doi:10.1021/acs.est.7b05864, 2018.

Kitanovski, Z., Hovorka, J., Kuta, J., Leoni, C., Prokeš, R., Sáňka, O., Shahpoury, P. and Lammel, G.: Nitrated monoaromatic hydrocarbons in ambient air – levels, mass size distributions and inhalation bioaccessibility, in preparation for Environ. Sci. Pollut. Res., 2020.

Marques, M.R.C., Loebenberg, R. and Almukainzi, M.: Simulated biological fluids with possible application in dissolution testing, Dissolution Technol., 15–28, doi:10.14227/DT180311P15, 2011.

Myrdal, P., Ward, G.H., Dannenfelser, R.M., Mishra, D., Yalkowsky, S.H.: AQUAFAC 1: Aqueous functional group activity coefficients; application to hydrocarbons, Chemosphere 24, 1047-1061, doi:10.1016/0045-6535(92)90196-X, 1992.

Schwarzenbach, R.P., Stierli, R., Folsom, B.R., Zyer, J.: Compound properties relevant for assessing the environmental partitioning of nitrophenols, Environ. Sci. Technol., 22, 83-92, doi:10.1021/es00166a009, 1988.

Shahpoury, P., Kitanovski, Z. and Lammel, G.: Snow scavenging and phase partitioning of nitrated and oxygenated aromatic hydrocarbons in polluted and remote environments in central Europe and the European Arctic, Atmos. Chem. Phys., 18, 13495–13510, doi:10.5194/acp-18-13495-2018, 2018.

9) Reviewer's comment:

Lines 331-333: I think similar mass size distributions of NMAHs and HULIS only indicate they may have similar atmospheric processes, and cannot indicate NC and NSAs are important constituents of HULIS.

*Authors' response:*

We agree with the comment. Corrections have been made in the text accordingly:

Page 12, Line 376-377: "*The MSDs of HULIS in MZ and TK closely followed the MSDs of NCs and NSAs (Figs. 1 and 2), suggesting that they may have undergone similar atmospheric processes.*"

10) Reviewer's comment:

Lines 260-261 and related analysis throughout the paper: How do the authors know the different concentration levels of the two sites are not due to different cut-off of the aerosol samples? I notice that both sites have the same cut-off of 0.49-0.95, 0.95-1.5, 1.5-3, 3-7.2 µm, it would be meaningful to compare the concentrations of the same particle size ranges and then analyse the different sources at the two sites.

*Authors' response:*

The comparison of the target compound concentrations in specific size ranges was already discussed in the original manuscript. However, in order to address the Reviewer's comment regarding sources and for consistency, we have now performed additional correlation analysis among different species in $PM_{0.95}$ and $PM_{3-0.95}$ at both sites (Tables S5, S6, S8, S9, S11, S12, S14 and S15), and provided additional discussion on the potential emission sources.

Page 8, Line 238- Page 10, Line 290: "*Initially, we did the correlation analysis on $PM_{10}$ (MZ) and total PM (TK) samples (Table S4 and S7). Although the correlation analysis was performed using limited number of sample sets per location (five for TK and four for MZ), it showed several interesting features. Based on these results, we propose the most probable sources ............................................. high correlations of 4-NC and MNC with $K^+$ in $PM_{0.95}$ indicate that BB was a significant emission source over MZ (Chow et al., 2016; Voliotis et al., 2017; Wang et al., 2018), whereas their high correlations with sulphate in $PM_{3-0.95}$ ($0.66 < R^2_{adj} < 1.00$; Table S9) could infer possible anthropogenic emissions, i.e. coal combustion (Lu et al., 2019a).*"

Page 11, Line 338- Page 12, Line 358: "*For this analysis, we considered the compositions of $PM_{10}$ (at MZ) and total PM (at TK), as well as the constituents of $PM_{0.97}$ and $PM_{3-0.97}$ at both sites. We found a significant correlation (n = 5, p<0.05) between $9,10-O_2ANT$ and $1,2-O_2BAA$ at TK site, regardless of the considered PM size range, which suggests a common ................................................ Finally, $K^+$, WSOC, and HULIS correlated significantly at TK (p<0.05, $R^2_{adj}$ 0.89-0.90), whereas such correlations were not found at MZ. In summary, while N/OPAHs from TK samples were influenced by primary emissions related to BB and fossil fuel combustion, those from MZ samples were dominated by aged air masses.*"

11) Reviewer's comment:

Some important information should be added in the Experimental section:

Section 2.2: Why sampling is conducted at the two sites? They are both urban sites. Please specify the differences of the two sites in section 2.2, such as referring previous studies in the two sites to address the differences. It is not quite clear for me what's the differences between the two sites even after finish reading the paper.

*Authors' response:*

Our aim was to study two different European cities dominated by different emission sources: Thessaloniki in south-eastern Europe, which is a biomass burning hotspot (Saffari et al., 2013; Velali et al., 2019) and Mainz, a central European city that is mainly affected by traffic emissions and long-range transport (Winkler & Junge, 1972; Wesp et al., 2000; Dusek et al., 2006). The following statement has been added to the text to address the Reviewer's comment:

Page 4, Line 101-104: "*These sites were selected to reflect the dominant emission sources in the study areas – while TK is a biomass burning hotspot in south eastern Europe (Saffari et al., 2013; Velali et al., 2019), MZ in central Europe is dominated by traffic emission and long-range transport (Winkler and Junge, 1972; Wesp et al., 2000; Dusek et al., 2006).*"

Dusek, U., Frank, G.P., Hildebrandt, L., Curtius, J., Schneider, J., Walter, S., Chand, D., Drewnick, F., Hings, S., Jung, D., Borrmann, S., Andreae, M.O.: Size matters more than chemistry for cloud nucleating ability of aerosol particles, Science, 312, 1375-1378, doi: 10.1126/science.1125261, 2006.

Saffari, A., Daher, N., Samara, C., Voutsa, D., Kouras, A., Manoli, E., Karagkiozidou, O., Vlachokostas, C., Moussiopoulos, N., Shafer, M.M., Schauer, J.J., Sioutas, C.: Increased Biomass Burning Due to the Economic Crisis in Greece and Its Adverse Impact on Wintertime Air Quality in Thessaloniki, Environ Sci Technol., 47, 13313-13320, doi:10.1021/es403847h, 2013.

Velali, E., Pantazaki, A., Besis, A., Choli-Papadopoulou, T., Samara, C.: Oxidative stress, DNA damage, and mutagenicity induced by the extractable organic matter of airborne particulates on bacterial models, Regul. Toxicol. Pharmacol., 104, 59-73, doi:10.1016/j.yrtph.2019.03.004, 2019.

Wesp, H.F., Tang, X., Edenharder, R.: The influence of automobile exhausts on mutagenicity of soils: contamination with, fractionation, separation, and preliminary identification of mutagens in the Salmonella/reversion assay and effects of solvent fractions on the sister-chromatid exchanges in human lymphocyte cultures and in the in vivo mouse bone marrow micronucleus assay, Mutation Res., 472, 1–21, doi:10.1016/S1383-5718(00)00088-7, 2000.

Winkler, P. and Junge, C.E.: Growth of atmospheric particles as a function of relative humidity, J. Rech. Atmos. 72, 617-638, 1972

12) Reviewer's comment:

Section 2.2 and Table 2: Only four sets of sample were collected at MZ site, and five sets at TK site. How can the authors confirm that only four or five samples can represent the conditions at the sites?

*Authors' response:*

In this study, we did not claim that sampling periods represent general winter conditions in MZ and TK sites. Samples were collected at both locations during the same winter season (2015/2016). In this season, emissions influencing the two sites are very different (please see our response to the previous comment). Representing the general winter conditions at these sites would require longer sampling periods and coverage of several years. Although with respect to temperature ranges and synoptically, the few weeks that our study covered at both sites were characterized by situations frequent in winter-time, i.e., south-westerly advection with moderate winds at MZ and weak southerly (on-shore) or north-easterly winds at TK (Saffari et al., 2013; Voliotis et al., 2017). We have added the following statement to the manuscript in order to clarify this point:

Page 4, Line 107-110: "*In this period, the emissions influencing the sample sites are very different and, in terms of temperature changes and synoptically, the sampling period is characterized by south-westerly advection with moderate winds at MZ and weak southerly or north-easterly winds at TK (Saffari et al., 2013; Voliotis et al., 2017).*"

Saffari, A., Daher, N., Samara, C., Voutsa, D., Kouras, A., Manoli, E., Karagkiozidou, O., Vlachokostas, C., Moussiopoulos, N., Shafer, M.M., Schauer, J.J., Sioutas, C.: Increased Biomass Burning Due to the Economic Crisis in Greece and Its Adverse Impact on Wintertime Air Quality in Thessaloniki, Environ Sci Technol., 47, 13313-13320, doi:10.1021/es403847h, 2013.

Voliotis A., Prokeš R., Lammel G., and Samara C.: New insights on humic-like substances associated with urban aerosols from central and southern Europe: size-resolved chemical characterization and optical properties. Atmos. Environ., 166, 286-299, doi:10.1016/j.atmosenv.2017.07.024, 2017.

**13) Reviewer's comment:**

Section 2.3: Please show the recoveries of the quantified species in Table 1. And please show the compound peaks and resolution of NMAHs in the LC-MS chromatograms.

*Authors' response:*

The following statements were added to the text to indicate the target analyte recoveries:

Page 6, Line 154-155: "*The mean recovery of target NMAHs at 100 pg µL$^{-1}$ was 91±3%.*"

Page 7, Line 187-188: "*The mean recoveries of target NPAHs and OPAHs at 200 pg µL$^{-1}$ were 73±15 and 72±18%, respectively.*"

Regarding chromatograms for NMAHs, we used the same method described by Kitanovski et al., 2012, and considering that this manuscript does not focus on analytical method development, we do not think that including the chromatograms in the manuscript would be appropriate. However, to address the Reviewer's comment, the chromatograms of NMAH standard mixture and a real sample from TK are shown at the end of this document.

**14) Reviewer's comment:**

The extraction and quantification of K+, WSOC and HULIS are omitted in the experimental section.

*Authors' response:*

This information was not included in the present manuscript since they are already presented in a companion paper (Voliotis et al. 2017). This was already mentioned at the end of the Introduction section. We have now moved the statement to Section 2.2.3:

Page 7, Line 203-205: "*The concentrations of ions, organic acids, HULIS and HULIS-C in the samples used in this study can be found in a companion paper (Voliotis et al., 2017)."*

Voliotis A., Prokeš R., Lammel G., and Samara C.: New insights on humic-like substances associated with urban aerosols from central and southern Europe: size-resolved chemical characterization and optical properties. Atmos. Environ., 166, 286-299, doi:10.1016/j.atmosenv.2017.07.024, 2017.

**15) Reviewer's comment:**

Besides, I suggest the authors to carefully check the data and analysis throughout the paper, and to draw scientific and validated conclusions. A better proofing reading is also needed.

*Authors' response:*

We thank the Reviewer for this suggestion. We have endeavoured to address this comment throughout the manuscript.

**16) Reviewer's comment:**

Specific comments:

Lines 71-73: This sentence is confusing. Which source is more dominated in urban areas? Primary sources or secondary formation? Please revise to be clear.

*Authors' response:*

We have revised the sentence as follows:

Page 3, Line 64-68: "*Nitrophenols (NPs), structurally related compounds to NCs, are emitted from primary sources (e.g. traffic, coal and wood combustion, industry and agricultural use of pesticides), which predominate their secondary formation in urban areas (Harrison et al., 2005; Cecinato et al., 2005; Hoffmann et al., 2007; Iinuma et al., 2007; Zhang et al., 2010; Ganranoo et al., 2010; Özel et al., 2011; Mohr et al., 2013; Kitanovski et al., 2012a and 2012b; Inomata et al., 2015; Teich et al., 2017; Wang et al., 2018; Lu et al., 2019a and 2019b).*"

17) Reviewer's comment:

Lines 103-105: I think these three references are more focused on the light absorption of nitro-aromatics. Could the authors add more references suggesting the light absorption capacity of PAHs and N/O-PAHs?

*Authors' response:*

We have added two new references regarding light absorption of PAHs and N/OPAHs:

Page 4, Line 93-95: "NMAHs, PAHs and N/OPAHs significantly contribute to the aerosol BrC due to their light- absorption capacity in the UV and visible range (Mohr et al., 2013; Samburova et al., 2016; Teich et al., 2017; Xie et al., 2017; Huang et al., 2018)."

Samburova, V., Connolly, J., Gyawali, M., Yatavelli, R.L.N., Watts, A.C., Chakrabarty, R.K., Zielinska, B., Moosmüller, H. and Khlystov, A.: Polycyclic aromatic hydrocarbons in biomass-burning emissions and their contribution to light absorption and aerosol toxicity, Sci. Total Environ., 568, 391-401, doi:10.1016/j.scitotenv.2016.06.026, 2016.

Huang, R.-J., Yang, L., Cao, J., Chen, Y., Chen, Q., Li, Y., Duan, J., Zhu, Ch., Dai, W., Wang, K., Lin, Ch., Ni, H., Corbin, J.C., Wu, Y., Zhang, R., Tie, X., Hoffmann, T., O'Dowd, C. and Dusek, U.: Brown carbon aerosol in urban Xi'an, Northwest China: The composition and light absorption properties, Environ. Sci. Technol., 52, 6825-6833, doi:10.1021/acs.est.8b02386, 2018.

18) Reviewer's comment:

Lines 309-401: Please clarify what are "these OPAHs" and what are "other N/OPAHs".

*Authors' response:*

The sentence has been revised as follow:

Page 11, Line 326-328: "*Such differences between size distributions indicate that 9-OFLN and 9,10-O$_2$ANT are subject to different atmospheric processes compared to all other N/OPAHs that we studied in the present work.*"

19) Reviewer's comment:

Lines 465-466: Could the authors further explain why larger particle size suggests aqueous-phase processes and slower formation?

*Authors' response:*

The following explanation was added to the text:

Page 14, Line 432-436: "*The activation of condensation mode particles (under high humidity) into cloud droplets, as well as the subsequent possible aqueous-phase reactions lead to the formation of larger particles in aged and cloud-processed aerosols. In central Europe, characteristic times of formation of coarse mode secondary inorganic aerosols and OC peak around 60-72 h (Lammel et al., 2003)*"

Lammel, G., Brüggemann, E., Gnauk, T., Müller, K., Neusüß, C., Röhrl, A.: A new method to study aerosol source contributions along the tracts of air parcels and its application to the near-ground level aerosol chemical composition in central Europe, J. Aerosol Sci., 34, 1-25, doi:10.1016/S0021-8502(02)00134-9, 2003.

[Figure]

LC-MS chromatogram of standard mixture of NMAHs at 100 pg µL$^{-1}$ level (EIC: Extracted Ion Chromatogram)

[Figure]

LC-MS chromatogram of PM$_{0.97-0.49}$ sample from Thessaloniki (17.03.2016)

**Anonymous Referee #2**

1) Reviewer's general comment:

The paper reports on findings from wintertime measurements of composition and mass size distribution of different nitrated and oxygenated aromatics in Mainz, Germany and Thessaloniki, Greece. Correlation coefficients between the concentration of these species and WSOC, HULIS, K+, and nitrate were determined to investigate sources of the observed N/O aromatics. The authors conclude that air masses sampled in Thessaloniki were impacted by fresh biomass burning while aged air masses (biomass and fossil fuel combustion) were sampled in Mainz. There is a lot of information on the total concentration of the various species and their size-dependent concentration, so there's certainly value to having this information for these two cities (despite the short duration of the measurements).

*Authors' response:*

We thank the Reviewer for their constructive comments, which we have addressed in detail. Please see our itemized responses below.

2) Reviewer's comment:

The way the paper is structured and the use of this many acronyms make the paper very hard to read.

*Authors' response:*

Following the Reviewer's comment, we have re-structured the manuscript 'Results' sections, as shown below:

3. Results and discussion
3.1 Sources of NMAHs and N/OPAHs at Thessaloniki and Mainz
3.1.1 Concentrations and sources of NMAHs
3.1.2 Concentrations and sources of N/OPAHs
3.2 Mass size distributions of target compounds
3.2.1 Mass size distributions of NMAHs
3.2.2 Mass size distributions of N/OPAHs
3.3 NMAHs as part of HULIS

Regarding the acronyms, we believe that removing the acronyms (replacing with actual substance names) would make the statements lengthy and difficult to follow for the readers, considering that our substances have relatively long names. Hence, we defined the abbreviations when they are first presented in the text, as well as in Table 1.

3) Reviewer's comment:

The other major comment I have is about the conclusions of the source attributions. For example, nitrate aerosols could be high in biomass burning plumes as well as aged urban plumes, so I'm not sure a correlation can be really conclusive.

*Authors' response:*

Following the comments from Reviewers 1 and 2, we have removed the discussions that relate target compound sources to the levels of nitrate.

4) Reviewer's comment:

Another support for the source apportionment conclusions is the mass size distributions; however, the resolution of these distributions is so low that I don't think they can be robust for such interpretation. The other comments are highlighted below. I recommend major revisions and reconsideration before accepting the paper for publication.

*Authors' response:*

In order to exceed the instrument detection limits, and for successful quantification of the target substances, we have used air samplers with the highest sampling rates; this comes with the down side of having lower number of impactor stages (i.e. relatively low size distribution). We fully understand the Reviewer's concerns and, to alleviate them, we have now removed the statements that link the compound mass size distribution to emission sources throughout Sections 3.1 and 3.2.

5) Reviewer's comment:

Line 9: what does the index in the summation sign indicate? It's probably the number of NMAHs, but perhaps it's more clear if it's defined for at least one group of compounds first.

*Authors' response:*

The index indicates the number of measured compounds (in this case NMAHs). The text has been changed as follows:

Page 1, Line 8-9: "*The total concentration of eleven NMAHs ($\sum_{11}$NMAH concentrations) measured in PM$_{10}$*"

6) Reviewer's comment:

Line 151: what's the time resolution of the samples in TK? I believe it flows better if section 2.2 is presented in the beginning of Section 2, followed by sample preparation and analytical methods. I also think section 2.1 (Chemicals and Solutions) can be moved to SI.

*Authors' response:*

The time resolution of the samples from TK is two days, as we noted in Table 2. We have added the following sentence to the current Section 2.1 (i.e. Collection of Samples) for further clarification:

Page 4, Line 115-116: "*For each sample set, the air was collected for the duration of 70 and 48 hours at MZ and TK, respectively.*"

Moreover, Section 2.1 is now moved to the Supplement (i.e. Section S1. Chemicals and solutions). Section 2 in the main text now begins with the "Section 2.1 Collection of samples".

7) Reviewer's comment:

L185: each filter paper or just sections of it?

*Authors' response:*

We have revised the related sentence as follows:

Page 6, Line 158-159: "*Briefly, two strips of each filter paper was placed inside a glass centrifuge tube...*"

8) Reviewer's comment:

L230: what justifies assuming that measurements at TK were also PM10? The authors later on do comment that perhaps larger than 10 um particles were sampled in TK (the statement on L 328- 329).

*Authors' response:*

It is true that the samples were collected without a $PM_{10}$ inlet at Thessaloniki (TK). We adopted an upper cut-off of 10 μm for TK samples *only* for the calculation of the mass median diameters (MMD) for consistency across sample sites (i.e. with Mainz samples). This may introduce a small underestimation of the MMD; however, during our sampling at TK, wind velocities did not favour re-suspension of giant particles or sea spray; this minimizes the contribution of PM larger than 10 μm. Although, the PM number size distribution was not measured in our study, at another European urban site (Ostrava, Czech Republic) in the same winter, also under prevailing weak winds, the measured $N_{<35\mu m}/N_{<10\mu m}$ was 1.0001 (Lammel et al., 2019). This ratio suggests that the underestimation of MMD is negligible under weak wind conditions. To address the Reviewer's comment, we have added the following statement to the manuscript:

Page 7, Line 199-203: "*For consistency across the samples, 0.001 μm was adopted as the lower cut-off of the lowermost stage (backup filter) and 10 μm as the upper cut-off of the uppermost stage, even in the absence of a $PM_{10}$ inlet in the case of TK samples. Although this may introduce small underestimation of MMDs for TK samples, during the sampling at TK, wind velocities did not favor resuspension of large particles and sea spray, hence, we expect that the contribution of PM >10 μm would be negligible.*"

Lammel, G., Kitanovski, Z., Kukučka, P., Novák, J., Arangio, A., Černikovsky, L, Codling, G.P., Filippi, A., Hovorka, J., Kuta, J., Leoni, C., Příbylová, P., Prokeš, R., Sáňka, O., Shahpoury, P., Tong, H.J. and Wietzoreck, M.: Levels, phase partitioning, mass size distributions and bioaccessibility of oxygenated and nitrated polycyclic aromatic hydrocarbons (OPAHs, NPAHs) in ambient air, Environ. Sci. Technol., submitted (2019)

9) Reviewer's comment:

L255-256: name of the country (Slovenia and China, etc) shouldn't be in () with the reference.

*Authors' response:*

The sentence has been revised as follows:

Page 8, Line 226-227: "*…$PM_{10}$ samples from Ljubljana, Slovenia (Kitanovski et al., 2012b) and Shanghai, China (Li et al., 2016).*"

**10)** Reviewer's comment:

L320: how can the contribution from primary traffic emissions explain the peak in MSD in the 0.95-1.5 um range? Primary emissions are typically peaking in <100 nm in number distribution, which puts the mass distribution peak at much smaller than 0.95-1.5 um.

*Authors' response:*

The sentence that the Reviewer is referring to has now been removed from the text.

**11)** Reviewer's comment:

L422: This sentence doesn't make sense. I thought NPYR is a marker for primary combustion; so why is "long range transported pollution" also included here? Despite this, the authors claim that lack of NPYR isomers suggests advection of chemically aged plumes to MZ. Aren't these sentences contradictory?

*Authors' response:*

We agree with the Reviewer's comment. The sentence has been rephrased as follows:

Page 11, Line 349 – Page 12, Line 358: "*For MZ $PM_{10}$ samples, we found significant correlations (n = 4, p<0.05) of 9-OFLN, BaOFLN, and 9-NANT with WSOC and HULIS (Table S13), without any significant correlations to $K^+$. We found similar correlations in $PM_{0.97}$, which suggest the predominance of chemically aged air masses that were advected during the MZ campaign. This is further supported by the absence of NPYR isomers in MZ samples, which are indicative of road traffic and industrial emissions (IARC, 1989; Finlayson-Pitts and Pitts, 2000; Lammel et al., 2017; Voliotis et al., 2017). Finally, $K^+$, WSOC, and HULIS correlated significantly at TK (p<0.05, $R^2_{adj}$ 0.89-0.90), whereas such correlations were not found at MZ. In summary, while N/OPAHs from TK samples were influenced by primary emissions related to BB and fossil fuel combustion, those from MZ samples were dominated by aged air masses.*"